

# Derivation of free energy, entropy and specific heat for planar Ising models: Application to Archimedean lattices and their duals

**Laurent Pierre[1⋆], Bernard Bernu[2†] and Laura Messio[2‡]**

**1** Université de Paris Nanterre
**2** Sorbonne Université, CNRS, Laboratoire de Physique Théorique
de la Matière Condensée, LPTMC, F-75005 Paris, France

⋆ lgrpierre@gmail.com , † bernard.bernu@sorbonne-universite.fr ,
‡ laura.messio@sorbonne-universite.fr

## Abstract

The 2d ferromagnetic Ising model was solved by Onsager on the square lattice in 1944, and an explicit expression of the free energy density $f$ is presently available for some other planar lattices. But determining exactly the critical temperature $T_c$ only requires a partial derivation of $f$. It has been performed on many lattices, including the 11 Archimedean lattices. In this article, we give general expressions of the free energy, energy, entropy and specific heat for planar lattices with a single type of non-crossing links. It is known that the specific heat exhibits a logarithmic singularity at $T_c$: $c_V(T) \sim -A\ln|1-T_c/T|$, in all the ferromagnetic and some antiferromagnetic cases. While the non-universal weight $A$ of the leading term has often been evaluated, this is not the case for the sub-leading order term $B$ such that $c_V(T)+A\ln|1-T_c/T| \sim B$, despite its significant impact on the $c_V(T)$ values in the vicinity of $T_c$, particularly important in experimental measurements. Explicit values of $T_c$, $A$, $B$ and other thermodynamic quantities are given for the Archimedean lattices and their duals for both ferromagnetic and antiferromagnetic interactions.



## Contents



# 1  Introduction

The Ising model is a simple model subject of extensive research since its introduction in 1920. Although it was solved early in one dimension (1d), the 2d case endured for many years, before several solutions were developed. The historical solution, found in 1944 by Onsager [1] on the square lattice, uses a transfer matrix and is a cumbersome calculation, later revisited by Kaufman [2]. Magnetization was then derived by Yang [3] in 1952. The second method, known as combinatorial, is due to Kac and Ward [4] and highlights the correspondence between the partition function and the count of loops, as formally derived in [5] and proved by Dolbilin et al [6]. Magnetization was then derived in this framework by Potts and Ward in 1955 [7]. Note that this method is related to the Pfaffian method used to solve the dimer problem [8, 9]. Finally, a Grassmann integral formulation was derived by Plechko [10, 11]. Relations between these various approaches have been discussed [12]. These methods are valid on any 2d lattice with non-crossing links. We refer to several articles reviewing this subject [13–15], and references therein. However, no exact solution has yet been derived in 3d, or in the presence of a magnetic field in 2d.

The exact methods described above confirm that all ferromagnetic Ising models on a periodic 2d lattice exhibit a phase transition in the thermodynamic limit, belonging to the same universality class, characterized by specific critical exponents. In particular, the critical exponent $\alpha = 0$ leads to a logarithmic singularity in the specific heat $c_V(T)$, with the principal behavior at the critical temperature $T_c$ given by: $c_V(\beta) \sim -A\ln|1 - \beta/\beta_c|$, where $\beta = 1/T$ and $\beta_c = 1/T_c$. The non-universal quantities $T_c$ and $A$ have been calculated on several lattices: the analytical expression of $T_c$ is known for all Archimedean and Laves lattices [16], while $A$ has been computed for the triangular, square and honeycomb lattices [15], as well as for several decorated lattices in [11]. However, to our knowledge, the subdominant term $B$ has never been previously determined except for four lattices by Gonzalez and the present authors in [17]:

$$c_V(\beta) = -A\ln\left|1 - \frac{\beta}{\beta_c}\right| + B + o(1). \tag{1}$$

$B$ is particularly important for comparing specific heat measurements near $T_c$ of experimental realizations of the 2d Ising model [17,18] to theoretical predictions, as well as for benchmarking extrapolation methods [17].

We will consider Archimedean lattices and their duals. Archimedean (also called uniform or semi-regular) tilings have equivalent vertices under symmetry operations (and so of the same degree $z$, called coordination number) and their faces are regular polygons. They include three regular tilings with equivalent faces (square, triangular and honeycomb tilings) and eight others, giving a total of eleven tilings, represented in Fig. 1. In this article, they are labeled (in the hexadecimal system) by the number of edges of the faces clockwise surrounding a site, with an A for Archimedean in front of these numbers. For example, A3CC is a tiling where each site belongs to a triangle and two dodecagons. The dual of a lattice is named with a D in front (for example, DA3CC is the dual of A3CC) and has a vertex on each face of the original lattice, with links between them if the corresponding faces share an edge. Note that DA4444 is A4444 and DA333333 is A666.

The first contribution of this article is an exhaustive and careful review of all the steps leading to the exact formula of the free energy per site for a planar Ising lattice through the combinatorial solution (with many details in the appendices). Special attention is paid to the treatment of the periodic boundary conditions and of the thermodynamic limit. Many articles discussing this method, its proof and its extensions focus on the square lattice. We emphasize that our proofs are valid for any planar lattice.

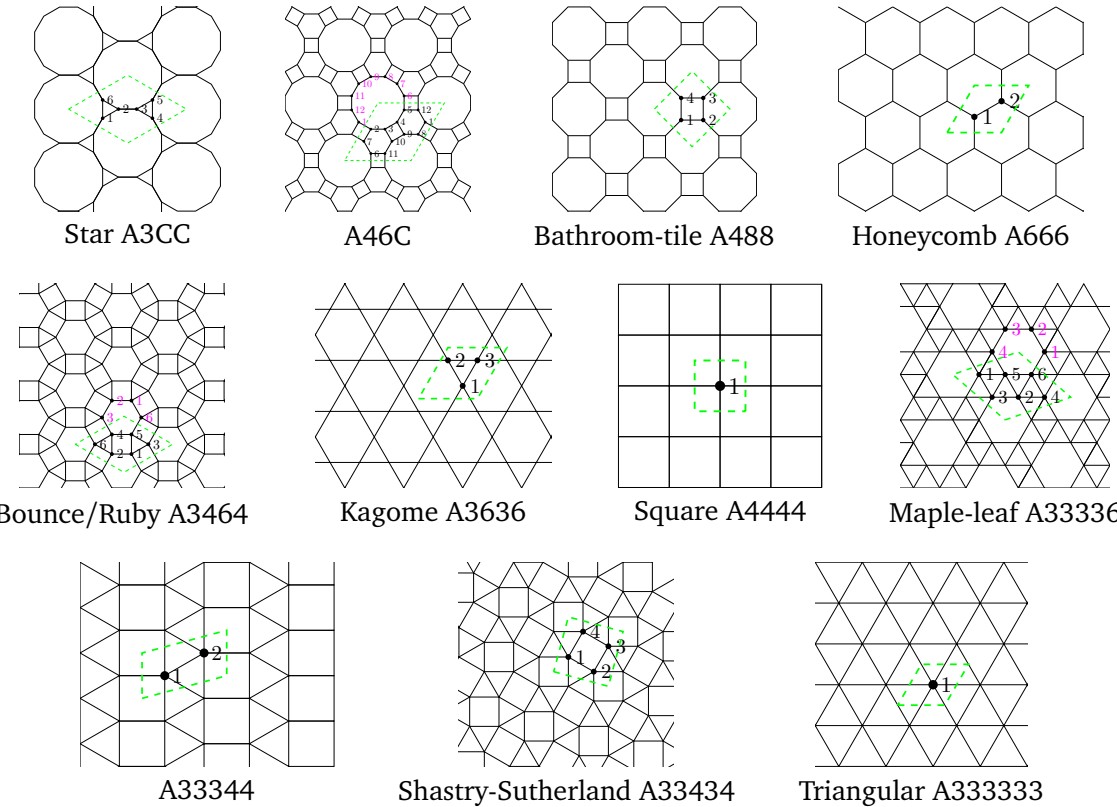

Figure 1: The 11 Archimedean tilings of the plane. The unit cell is indicated, and a possible choice of $m$ translationally inequivalent sites.

The second contribution is a complete exploitation of the combinatorial solution to derive the expression (either analytical or numerical) of $A$ and $B$, as well as thermodynamic quantities characterizing the transition and the ground state (with again all the details relegated to the appendices).

This article begins by revisiting the derivation of the free energy per site $f$ of a planar Ising model using the combinatorial solution, valid for both ferromagnetic (F) and antiferromagnetic (AF) models (Sec. 2), and applying it to Archimedean lattices in Section 3. In Section 4 are derived the expressions of the energy $e$, the entropy $s$, the specific heat $c_V$ per site, and the coefficients $A$ and $B$, while Sec. 5 recovers ground state properties $e_0$ and $s_0$ for both ferromagnetic (F) and antiferromagnetic (AF) models. In Sec. 6 we exploit two well known ways to relate the partition function of different lattices, which are duality and star-triangle transformation. Thus, we extend our results to the dual Archimedean lattices: the Laves lattices, and establish relations between several lattices. Finally, the conclusion and discussion are presented in Section 7.

## 2 The combinatorial solution

### 2.1 Kac Ward identity for a planar graph

The aim is to calculate the free energy per site $f(\beta)$ of an Ising model on a planar lattice, in the thermodynamic limit. In graph theory a graph is planar if it can be drawn on a plane with links (or edges), which are non intersecting curves. Our definition is more restrictive: A planar

graph is actually drawn on a plane and its links are non intersecting straight line segments.

Let $\mathcal{L}$ be a finite planar graph (hence without periodic boundary conditions, which are relegated to further discussion) of $N_l = \#\mathcal{L}$ undirected links and $N_s$ sites. The Ising variables on each site $i$ are $\sigma_i = \pm 1$. The energy reads:

$$E = -\sum_{\langle i,j \rangle \in \mathcal{L}} J_{ij} \sigma_i \sigma_j. \tag{2}$$

We define:

$$v_{ij} = \tanh \beta J_{ij}. \tag{3}$$

From the identity $e^{\pm a} = \cosh a \, (1 \pm \tanh a)$, the partition function simply writes:

$$Z = \sum_{\{\sigma_i = \pm 1\}} e^{\beta \sum_{\langle i,j \rangle} J_{ij} \sigma_i \sigma_j} \tag{4}$$

$$= \left( \prod_{\langle i,j \rangle} (1 - v_{ij}^2) \right)^{-1/2} \sum_{\{\sigma_i = \pm 1\}} \prod_{\langle i,j \rangle} (1 + v_{ij} \sigma_i \sigma_j). \tag{5}$$

From now on, we assume that all $v_{ij}$ have the same value

$$v = \tanh \beta J. \tag{6}$$

For ferromagnetic interactions ($J > 0$), $v$ varies from 0 (at high temperature) to 1 (at $T = 0$). For antiferromagnetic interactions ($J < 0$), $v$ varies from 0 to $-1$. In Eq. (5), each term of the expanded product can be associated with a subgraph $G$ of the graph $\mathcal{L}$ and is $v^{\#G} \prod_i \sigma_i^{d_i}$. The degree $d_i$ is the number of links in $G$ that include site $i$ and $\#G$ is the number of links in $G$. If at least one $d_i$ is odd, the contribution of this term is zero when summed over all spin configurations. Therefore, only subgraphs where all $d_i$ are even contribute, and these are called even subgraphs. Summing over all spin configurations, the contribution of an even subgraph gives a term $2^{N_s} v^{\#G}$. With $g_r$ the number of even subgraphs with $r$ links, we obtain:

$$Z = 2^{N_s} (1 - v^2)^{-N_l/2} \sum_{r=0}^{N_l} v^r g_r. \tag{7}$$

This was used by Hendrik Kramers and Gregory Wannier in 1941 [19] to relate $Z$ to the partition function $\tilde{Z}$ of the dual graph (see Sec. 6.1).

The combinatorial method uses the last equation (7) and the following Kac-Ward's identity:

$$\sum_{r=0}^{N_l} v^r g_r = \sqrt{\det(I_{2N_l} - v\Lambda)}, \tag{8}$$

where $I_{2N_l}$ is the identity matrix of size $2N_l$ and $\Lambda$ is a square matrix of the same size, whose rows and columns are labeled by the directed links of $\mathcal{L}$. A directed link $l = l_i \rightarrow l_f$ is characterized by an angle $\alpha_l$ with a reference direction. We denote $l_i$ and $l_f$ its initial and final sites (see Fig. 2a).

The $\Lambda$-coefficients are:

$$\Lambda_{l,l'} = \delta_{l_f l'_i} (1 - \delta_{l_i l'_f}) e^{i[\alpha_{l'} - \alpha_l]/2}, \tag{9}$$

where the brackets $[.]$ mean that the sum is reduced to the interval $]-\pi, \pi]$ modulo $2\pi$. The Kronecker symbols $\delta$ ensure that $l$ and $l'$ can be successive oriented links of a non back-tracking

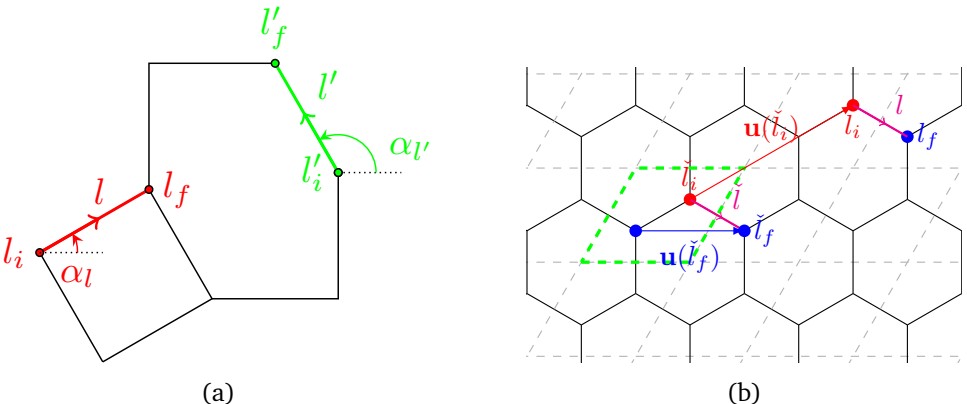

Figure 2: Definition of some quantities related (a) to any lattice and (b) to a periodic lattice. The dashed green contour is around the basic unit cell, with $m = 2$ sites in this example.

path. The matrix $I_{2N_l} - \nu\Lambda$ is the so-called Kac-Ward matrix [4,6,20]. From Eq. (8), the Kac-Ward determinant $P(\nu) \overset{\text{def}}{=} \det(I_{2N_l} - \nu\Lambda)$ has to be the square of a polynomial in $\nu$ of degree at most $N_l$. Eq. (8) is proved in a simple way by Lis [21], or by the Kasteleyn method [8] that maps the Ising model to a dimer tiling on an auxiliary graph with directed links. These two methods were not simply related until recently [12], through the use of a so-called *terminal graph*. In App. A, we derive a proof of this formula, partly similar to [21], but without requiring to transform the graph into a trivalent graph. Note that this proof is done in the more general case of link-dependent $\nu_l$. In this proof, the first stage A.1 proves that Kac-Ward determinant $P$ is the square of a polynomial denoted $\sqrt{P}$. But this could be achieved by showing that $P$ is also the determinant of an antisymmetric matrix (based on terminal graph) and defining $\sqrt{P}$ as its Pfaffian.

## 2.2 Finite and infinite periodic lattices

$\mathcal{L}$ is now a periodic lattice. It is drawn on a flat torus, containing $N_{uc}$ unit cells, $N_s$ sites and $N_l$ links. We define $m = N_s/N_{uc}$, $n_l = N_l/N_{uc}$ and $\tilde{m} = n_l - m$, the numbers of sites, links and faces per unit cell. After recalling how to adapt the Kac-Ward identity of Eq. (8) to a torus, we will give a simple formula that is valid in the thermodynamic limit.

Although $\mathcal{L}$ is not planar, nothing forbids to naively extend the definition of $\Lambda$, taking in Eq. (9) the lattice angles on a flat torus. The coefficient $\Lambda_{l,l'}$ is unchanged by translation of both $l$ and $l'$ by a lattice vector. It allows us to consider $\hat{\Lambda}$, the Fourier transform of $\Lambda$, with $N_{uc}$ diagonal blocks $\hat{W}(\mathbf{k})$ of size $2n_l \times 2n_l$, with $\mathbf{k} = (k_x, k_y)$ a vector of the Brillouin zone (BZ). Let $\mathbf{u}(s)$ denote the Bravais lattice vector associated to a site $s$ such that site $s - \mathbf{u}(s)$ is the translate of $s$ into the basic unit cell and $\breve{l} \overset{\text{def}}{=} (l_i - \mathbf{u}(l_i)) \to (l_f - \mathbf{u}(l_i))$ the translate of an oriented link $l$, with a new initial site into the basic unit cell: $\mathbf{u}(\breve{l}_i) = 0$, see Fig. 2b. The matrix $\hat{W}(\mathbf{k})$ is defined as:

$$\hat{W}(\mathbf{k})_{\breve{l},\breve{l}'} = e^{i\mathbf{k}\cdot\mathbf{u}(\breve{l}_f)}\Lambda_{\breve{l},\breve{l}'+\mathbf{u}(\breve{l}_f)}. \tag{10}$$

Note that $\mathbf{u}(\breve{l}_f) = \mathbf{u}(l_f) - \mathbf{u}(l_i)$ is non-zero when the sites of $\breve{l}$ are in different unit cells, as in the example of Fig. 2b. The notation $\breve{l}$ will no more be used for the indices of $\hat{W}(\mathbf{k})$ as there is no possible ambiguity with $l$. We also define:

$$P_{\mathbf{k}}(\nu) = \det(I_{2n_l} - \nu\hat{W}(\mathbf{k})). \tag{11}$$

Note that the transformation $\Lambda \mapsto \hat{\Lambda}$ is such that $\det(I_{2N_l} - v\Lambda) = \det(I_{2N_l} - v\hat{\Lambda}) = \prod_{\mathbf{k}} P_{\mathbf{k}}(v)$, where the product is over all the BZ wavevectors (it would have been a unitary transformation by replacing $\mathbf{u}(\check{l}_f)$ by $\left(\mathbf{u}(\check{l}_f) - \mathbf{u}(\check{l}'_f)\right)/2$ in Eq. (10)).

Since a graph drawn on a torus is not planar, the Kac Ward identity (8) has to be adapted in this case. Any such graph can be seen as a minimal periodic lattice, with a single unit cell of $N_s$ sites, for which Eq. (10) is meaningful. The adaptation of the Kac Ward identity (see App. B) then involves four polynomials which are square roots of determinants (proof in App. B.1):

$$\sum_{r=0}^{N_l} v^r g_r = \frac{1}{2} \left( \sqrt{P_{(0,\pi)}} + \sqrt{P_{(\pi,0)}} + \sqrt{P_{(\pi,\pi)}} - \sqrt{P_{(0,0)}} \right)(v). \tag{12}$$

The polynomial $P_{\mathbf{k}}$ is only used here for the 4 special reciprocal vectors $\mathbf{k}$ such that $2\mathbf{k} = 0$. The single unit cell of $N_l$ links of the graph is considered with periodic ($k_x = 0$ or $k_y = 0$) or antiperiodic ($k_x = \pi$ or $k_y = \pi$) boundary conditions, and $\hat{W}(\mathbf{k})$ is simply $\Lambda$ in which coefficients of links crossing the antiperiodic boundaries are multiplied by $-1$. The proof of Eq. (12) is recalled in App. B.2.

Similarly, it is possible to calculate the partition function $Z$ exactly on an orientable surface of genus $g$. Then $Z$ is a weighted average of $4^g$ polynomials which are square roots of determinants [7, 22, 23] (see App. B.3).

For a periodic lattice on a torus, with $\omega \times \omega$ unit cells, $P_{\mathbf{k}}$ is the determinant of a matrix of size $2\omega^2 n_l \times 2\omega^2 n_l$. Each of the four occurences of $P_{\mathbf{k}}$ in Eq. (12), can be Fourier transformed. Then Eq. (12) writes in terms of the $P_{\mathbf{k}}$ of a unit cell torus ($\omega = 1$):

$$\sum_{r=0}^{\omega^2 n_l} v^r g_r = \frac{1}{2} \left( \sqrt{\prod_{\mathbf{k}} P_{\mathbf{k}+(0,\frac{\pi}{\omega})}} + \sqrt{\prod_{\mathbf{k}} P_{\mathbf{k}+(\frac{\pi}{\omega},0)}} + \sqrt{\prod_{\mathbf{k}} P_{\mathbf{k}+(\frac{\pi}{\omega},\frac{\pi}{\omega})}} - \sqrt{\prod_{\mathbf{k}} P_{\mathbf{k}}} \right)(v), \tag{13}$$

where $k$ assumes the $\omega^2$ values $\left(\frac{2\pi}{\omega} i_x, \frac{2\pi}{\omega} i_y\right)$ of the reciprocal space, with $i_x, i_y = 0 \ldots \omega - 1$.

We know from the proof of Eq. (12) (in App. B.1) that each product $\prod_{\mathbf{k}'} P_{\mathbf{k}'}$ is the square of a polynomial. However, it is not the case for each of its factors: most are pairwise equal, since $P_{-k'} = P_{k'}$ and only the four unpaired factors are squares, since then $\mathbf{k}' = -\mathbf{k}'$, or equivalently $2\mathbf{k}' = 0$, and we fall in the situation of Eq. (12). Hence, we cannot exchange the $\sqrt{\phantom{x}}$ and $\prod$ symbols, but the four products are definitely squared polynomials in $v$ and the four square roots are polynomials of constant term 1.

The four products of Eq. (13) are equivalent in the thermodynamic limit and

$$\lim_{\omega \to \infty} \left( \sum_{r=0}^{\omega^2 n_l} v^r g_r \right)^{1/\omega^2},$$

is the geometric mean of $\sqrt{P_{\mathbf{k}}(v)}$ over the BZ. Hence the free energy density $f = -T \ln Z / N_s$ in the thermodynamic limit is correct if we naively extend the definition of $\Lambda$ to periodic lattices, taking in (9) the lattice angles on a flat torus. Thus the thermodynamic limit reads (for all $|v| < 1$):

$$-\beta m f = m \ln 2 - \frac{n_l}{2} \ln(1 - v^2) + \frac{1}{2} \int_{BZ} \frac{d^2 \mathbf{k}}{4\pi^2} \ln P_{\mathbf{k}}(v). \tag{14}$$

Note that the integral is the average of $\ln P_{\mathbf{k}}(v)$ over the BZ of area $4\pi^2$.

## 2.3 Calculation of $P_{\mathbf{k}}(v)$

Knowing $P_0$ among all $P_{\mathbf{k}}$ is sufficient to obtain $\beta_c$ for F models (see the end of Sec. 2.5). However, knowing $P_{\mathbf{k}}$ over the whole BZ gives much more information than just $P_0$: indeed

---

**Algorithm 1** Faddeev-Le Verrier algorithm

---

$M = I_n$
$p_0 = 1$
**for** $j$ from 1 to $n$ **do**
  $M \mathbin{*}= \hat{W}(\mathbf{k})$
  $p_j = -(\operatorname{Tr} M)/j$
  $M \mathbin{+}= p_j * I_n$
**end for**

---

Eq. (14) allows to determine the quantities $A$ and $B$ of Eq. (1), as well as the residual entropy and energy of non-ordered AF models down to $T = 0$.

The coefficients of the polynomial $P_{\mathbf{k}} = \sum_j p_j v^j$ are also the coefficients (in reverse order) of $\sum_j p_j v^{n-j} = v^n P_{\mathbf{k}}(1/v) = \det(v I_n - \hat{W}(\mathbf{k}))$, which is the characteristic polynomial of the matrix $\hat{W}(\mathbf{k})$. Hence they are given by the Faddeev-Le Verrier algorithm, see Algo. 1. Note that $n = 2n_l$ is the size of the matrix $\hat{W}(\mathbf{k})$, and the degree of its characteristic polynomial, but the degree of $P_k$ may be lower than $n$ (see Sec. 2.5).

App. C.1 explains how to replace every non null coefficient of the matrix $\hat{W}(\mathbf{k})$ with a mononial of $\phi$, $\varphi$, $1/\phi$ and $1/\varphi$, where $\phi = e^{ik_x}$ and $\varphi = e^{ik_y}$. Then the Faddeev-Le Verrier algorithm handles only integer Laurent polynomials of low degrees in $\phi$ and $\varphi$. It is very efficient and may reduce to less than $4n^5 z$ additions of integers, where $z$ is the maximal degree of sites (and the coordination number for an Archimedean Lattice). This method can be adapted when the links in the unit cell do not all have the same $J_l$. If the various $v_l$ assume only $k$ different values $v_1, \ldots, v_k$, the Fadeev-Le Verrier algorithm will work on the matrix $\hat{W}(\mathbf{k})v/v_1$, where $v$ is now the diagonal matrix of coefficients $v_l$. It will handle polynomials of low degree in $\phi$, $\varphi$, $1/\phi$, $1/\varphi$, $r_2$, ... $r_k$ where $r_j = v_j/v_1$. The computation time is multiplied by less than $n^{k-1}$.

## 2.4  Some properties of the polynomial $P(v)$ for a planar graph

We denote $\bar{P} = \sqrt{P}$. Its constant term is 1. According to Eq. (8), $\deg \bar{P}$ is the number of links of the largest even subgraph (i.e. with all sites of even degree).

A planar graph $\mathcal{L}$ has a dual graph $\tilde{\mathcal{L}}$ in which vertices become faces, faces become vertices and edges are rotated by about a quarter turn (see Fig. 3). Duality has been studied intensively because $\bar{P}(v)$ is proportional to $(1+v)^{N_l} \bar{\tilde{P}}\left(\frac{1-v}{1+v}\right)$ and the square lattice is self-dual (see Sec. 6.1). Here we only use this proportionality to infer that the multiplicity of the factor $1+v$ in $\bar{P}$ is $N_l - \deg \bar{\tilde{P}}$. It is the minimal number of links to remove to make the dual graph even. It is also (since a face with $h$ edges generates a site of degree $h$ on the dual lattice) the minimal number of links to remove to make all the faces even, or (since the graph is planar) the minimal number of links to remove to make the graph bipartite, or (since all $J$'s are equal) the number of frustrated links in the AF ground state.

When the graph is drawn on a torus, it is not planar and we cannot use the dual graph to prove it, but the multiplicity of the factor $1 + v$ in $Z$ is still the number of frustrated links in the AF ground state, since an extra factor $1+v$ in $Z$ increases the free energy by $\ln(1+v)$ and the energy of the AF ground state ($v = -1$) by $2|J|$.

## 2.5  Properties of the polynomial $P_{\mathbf{k}}(v)$

When $2\mathbf{k} = \mathbf{0}$, $P_{\mathbf{k}}$ is the square of a polynomial $\sqrt{P_{\mathbf{k}}}$ of constant term 1 denoted in the following $\bar{P}_{\mathbf{k}}$. This ensures that the right-hand side of Eq. (8) (special case where no link crosses the

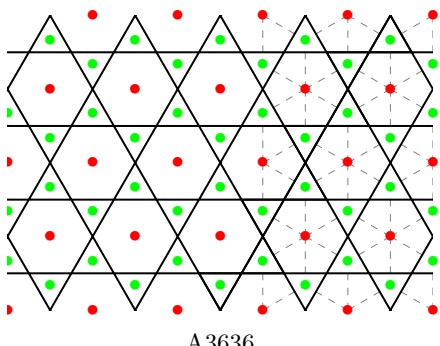
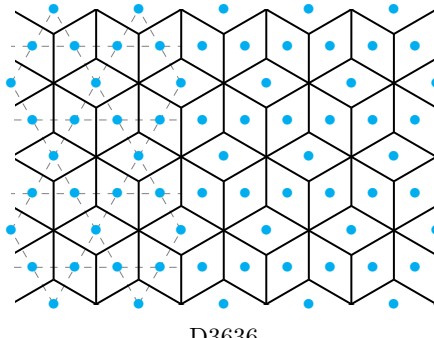

Figure 3: Duality relation between A3636 and D36363. Red and green points are both the center of hexagonal and triangular faces of A3636 and the vertices of D36363. Reciprocally, cyan points are the center of the losange faces of D3636 and the vertices of A3636.

boundary) and the four terms in the right-hand side of Eq. (12) are polynomials.

A first set of properties will concern the multiplicity of the factor $1+v$ or $v$ in various polynomials, requiring the definition of several lattice-dependant numbers:

- $N_v$ is the number of links to remove in the one-cell torus to make it bipartite.

- $N_e$ is the number of links to remove in the one-cell torus to make all faces with an even number of links, or equivalently to make its dual even.

- $n_v$ is the minimal average number of links to remove per unit cell to make the lattice bipartite, or with even faces. It is the same number since the lattice is planar.

Obviously $N_v \geq N_e \geq n_v$. For Archimedean lattices, $n_v$ is an integer, but not necessarily for other lattices. $N_v - n_v$ may be large. In Fig. 4, two lattices are given where $N_e = 2n_v = 1$ or 6. On A4444, $N_v = 2 > N_e = n_v = 0$. These numbers give the multiplicity of the factor $1+v$ in some polynomials:

- There are $N_v$ frustrated links at $T = 0$ for the AF one-cell torus. So $N_v$ is the multiplicity of the factor $1+v$ in its partition function and in $\bar{P}_{(0,\pi)} + \bar{P}_{(\pi,0)} + \bar{P}_{(\pi,\pi)} - \bar{P}_{(0,0)}$.

- If $2\mathbf{k} = 0$ then $\deg \bar{\bar{P}}_{\mathbf{k}} \leq n_l - N_e$ and $(1 + v)^{N_e}$ divides $\bar{P}_{\mathbf{k}}$.

- $P_{\mathbf{k}}(v)$ is an integer Laurent polynomial in $v$, $\phi = e^{ik_x}$ and $\varphi = e^{ik_y}$. Let $\text{val}_{1+v}(P_{\mathbf{k}}(v))$ be the multiplicity of the factor $1+v$ in this polynomial. Then the multiplicity of $1+v$ in the products $\prod_{\mathbf{k}'} P_{\mathbf{k}'}$ in Eq. (13) is proved equivalent to $\omega^2 \text{val}_{1+v}(P_{\mathbf{k}}(v))$ in App. C.2. But the multiplicity of $1+v$ in the polynomials $\sqrt{\prod_{\mathbf{k}'} P_{\mathbf{k}'}}$ is the number of frustrated links, which is equivalent to $\omega^2 n_v$. Hence $2n_v = \text{val}_{1+v}(P_{\mathbf{k}}(v))$. This proves that $n_v$ is an integer or half an integer.

Other properties are:

- $P_{\mathbf{k}} - P_0$ is divisible by $v^{n_a}$, where $n_a > 0$ is the minimal length of a loop with a non zero winding number in the one cell torus (see App. C.3).

- For Archimedean lattices, the degree of $P_{\mathbf{k}}$ is $2m\lfloor z/2 \rfloor$, where $z = n/m$ is the coordination number i.e. the common degree of all sites and $\lfloor . \rfloor$ the floor function. More generally we prove in App. C.4 that $\deg P_k \leq 2 \sum_a \lfloor c_a/2 \rfloor$ where $c_a$ is the degree of site $a$. As well,

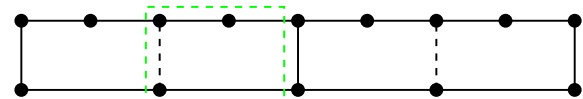

Unbalanced ladder, $N_v = N_e = 2n_v = 1$.

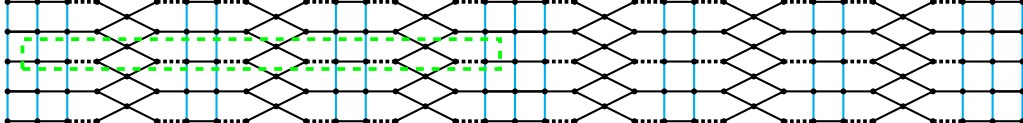

This lattice has 16 sites per unit cell and $N_v = 7 > N_e = 2n_v = 6$.

Figure 4: Two sample lattices where $N_e = 2n_v > 0$. For a torus made of the single cell surrounded by the green dashed line, we have to remove the blue links to make the graph bipartite ($N_v$ links), and the black dashed links to make the faces even ($N_e$ links). But to make the infinite lattice bipartite, we need only to remove the black dashed links in every other cell ($n_v$ links per unit cell in average).

the degree of $\bar{P}_k$ is $m\lfloor z/2 \rfloor$ when $2k = 0$ since non-null coefficients correspond to even subgraphs (see A.3). For example, the honeycomb lattice (A666) has $2n_l = 6$ but the degree of $P_k$ is only $2m\lfloor z/2 \rfloor = 4$.

- When $2\mathbf{k} = \mathbf{0}$, $\bar{P}_k$ takes special values at 0 (infinite temperature) and 1 (F ground state):

$$\bar{P}_0(0) = \bar{P}_{(0,\pi)}(0) = \bar{P}_{(\pi,\pi)}(0) = \bar{P}_{(\pi,0)}(0) = 1\,, \tag{15a}$$

$$-\bar{P}_0(1) = \bar{P}_{(0,\pi)}(1) = \bar{P}_{(\pi,\pi)}(1) = \bar{P}_{(\pi,0)}(1) = 2^{\tilde{m}} = 2^{n_l - m}\,. \tag{15b}$$

Eq. (15a) is proved in App. B.1 and Eq. (15b) in App. C.5. This proves the existence of a finite temperature phase transition in any 2d ferromagnetic model as $\bar{P}_0(v)$ cancels between $v = 0$ and $v = 1$, which leads to a singularity in Eq. (14).

We define the functions $\xi_i(\mathbf{k})$, $X_\mathbf{k}(v)$, $a_i(v)$ and the integers $n_f$, $\mathbf{c}_{ij}$, $n_i$ such that:

$$X_\mathbf{k}(v) = \frac{P_\mathbf{k}(v)}{(1 + v)^{2n_v}} = a_0^2(v) + \sum_{i=1}^{n_f} a_i(v)\, \xi_i(\mathbf{k})\,, \tag{16}$$

$$\xi_i(\mathbf{k}) = \sum_{j=1}^{n_i} \sin^2 \frac{\mathbf{c}_{ij} \cdot \mathbf{k}}{2}\,. \tag{17}$$

The functions $a_0^2$ and $a_{i>0}$ are polynomials. From these definitions, we deduce the following properties:

- $a_0(0) = 1$, since we choose $a_0 = +(1 + v)^{-n_v} \bar{P}_0$.

- Since $\bar{P}_0(1) = -2^{n_l - m}$ (see Eq. (15b)), we have:

$$a_0(1) = -2^{n_l - m - n_v}\,. \tag{18}$$

- The polynomials $a_{i>0}(v)$ are divisible by $v^{n_a}$, and thus $a_i(0) = 0$.

Finally, from Eq. (14) and the definitions above, the free energy density $f$ takes the form:

$$-\beta m f = -\beta m f_0 + \frac{1}{2} \int_{BZ} \frac{d^2\mathbf{k}}{4\pi^2} \ln X_\mathbf{k}(v)\,, \tag{19}$$

with

$$-\beta m f_0 = m \ln 2 - \frac{n_l}{2} \ln(1 - v^2) + n_v \ln(1 + v). \tag{20}$$

The free energy density $f$ is a continuous function of the temperature, thus the integral in Eq. (14) or Eq. (19) converges. But singularities may occur when $X_{\mathbf{k}}(v)$ vanishes for some $\mathbf{k}$ and $v$ (or equivalently when $P_{\mathbf{k}}(v)$ vanishes). This happens in the F models for $\mathbf{k} = \mathbf{0}$, as $a_0$ has a root $v_c$ and $\xi_i(\mathbf{0}) = 0$, leading to a critical inverse temperature $\beta_c$ verifying

$$J\beta_c = \operatorname{atanh}(v_c). \tag{21}$$

For non-F models (either AF or models with both positive and negative values of exchanges), zero, one or even more singularities can exist, for $\mathbf{k} = \mathbf{0}$ or $\mathbf{k} \neq \mathbf{0}$. An instructive example is given in [24], where three successive phase transitions occur on a $J_1 - J_2$ Shastry-Sutherland model (reentrancy of an ordered phase).

# 3 Partition function of the Archimedean and Laves lattices

We recall that all the sites of an Archimedean lattice have the same degree $z = 2n_l/m$, while in its dual, also called Laves lattice, all the faces have the same number of sides. Tab. 1 gives some characteristics of these lattices, while the polynomial $X_{\mathbf{k}}(v)$ defined in Eq. (16) and fully characterizing the free energy density (19) is given in Tab. 2. The code computing $X_{\mathbf{k}}(v)$ for the Archimedean lattices is provided in the Supp. Mat. [25].

Table 1: Characteristic quantities of the Archimedean lattices and of their dual lattices. We have the coordination number $z$, the number $m$ of sites per unit cell, the number $n_l$ of links per unit cell and the multiplicity $2n_v$ of the factor $1 + v$ in $P_{\mathbf{k}}(v)$. The critical value $v_c$ of the F model is related to the critical temperature through Eq. (6). The integer $n_f$ is defined in Eq. (16). $F(a) = g\left(\sqrt[3]{37 + 27\sqrt{a} + 3\sqrt{3(51 + 27a + 74\sqrt{a})}}\right)$ and $g(x) = (x^2 - 2x - 2)/(x^2 + 4x - 2)$ (exact formula in [16]).

| lattice | $z$ | $m$ | $n_l$ | $n_v$ | $\tilde{m}$ | $\tilde{n}_v$ | $n_f$ | $v_c$ | First determination |
|---|---|---|---|---|---|---|---|---|---|
| A3CC | 3 | 6 | 9 | 2 | 3 | 3 | 1 | $\frac{\sqrt{12+10\sqrt{3}}-1-\sqrt{3}}{4}$ | 1972 [26] |
| A46C | 3 | 12 | 18 | 0 | 6 | 6 | 3 | $\sqrt{\frac{5+3\sqrt{3}-\sqrt{44+26\sqrt{3}}}{2}}$ | 1985 [27, 28] |
| A488 | 3 | 4 | 6 | 0 | 2 | 2 | 2 | $\frac{\sqrt{10+8\sqrt{2}}-2-\sqrt{2}}{2}$ | 1951 [29] |
| A666 | 3 | 2 | 3 | 0 | 1 | 1 | 1 | $\frac{1}{\sqrt{3}}$ | 1944 [1] |
| A3464 | 4 | 6 | 12 | 2 | 6 | 0 | 3 | $\frac{1+\sqrt{3}-\sqrt[4]{12}}{2}$ | 1983 [28, 30] |
| A3636 | 4 | 3 | 6 | 2 | 3 | 0 | 1 | $\frac{1+\sqrt{3}-\sqrt[4]{12}}{2}$ | 1951 [31] |
| A4444 | 4 | 1 | 2 | 0 | 1 | 0 | 1 | $\sqrt{2}-1$ | 1941 [19] |
| A33336 | 5 | 6 | 15 | 4 | 9 | 3 | 3 | $F(3)$ | 2010 [28] |
| A33344 | 5 | 2 | 5 | 1 | 3 | 1 | 3 | $1/3$ | 1974 [32] |
| A33434 | 5 | 4 | 10 | 2 | 6 | 2 | 3 | $F(2)$ | 1974 [32] |
| A333333 | 6 | 1 | 3 | 1 | 2 | 0 | 1 | $2-\sqrt{3}$ | 1945 [33] |

Table 2: Polynomials of $X_{\mathbf{k}}(v)$ (see Eq. (16)) used in the free energy density Eq. (19) for the Archimedean lattices. The double brakets stand for $[\![\mathbf{c}_{i1}\ \mathbf{c}_{i2}\ ...\mathbf{c}_{in}]\!] = \xi_i(\mathbf{k}) = \sum_{j=1}^{n_i}\sin^2\frac{\mathbf{c}_{ij}\cdot\mathbf{k}}{2}$. In $a_3(v)$ of A3464's model: $w = v + 1/v$. The way to obtain these polynomials for the dual lattices is given in App. G.

| lattice | $a_0(v)$ | $a_i(v)$ | $\xi_i(\mathbf{k})$ |
|---|---|---|---|
| A3CC | $(1-v+v^2)^2-3v^4$ | $4v^4(1-v)(1-v+2v^2)$ | $\left[\begin{smallmatrix}1&0&1\\0&1&1\end{smallmatrix}\right]$ |
| A46C | $(1+2v^2+5v^4)\times$ $((v^4-5v^2+2)^2-3(3v^2-1)^2)$ | $-4v^{12}(1-v^2)^6$ | $\left[\begin{smallmatrix}2&0&2\\0&2&2\end{smallmatrix}\right]$ |
|  |  | $8v^{10}(1-v^2)^4(2+3v^2+4v^4-v^6)$ | $\left[\begin{smallmatrix}1&2&1\\2&1&-1\end{smallmatrix}\right]$ |
|  |  | $8v^6(1-v^2)^2(2v^4+v^2+1)\times$ $(5v^{10}+38v^8+25v^6+19v^4+6v^2+3)$ | $\left[\begin{smallmatrix}1&0&1\\0&1&1\end{smallmatrix}\right]$ |
| A488 | $1-4v^3-v^4$ | $8v^3(1-v^4)$ | $\left[\begin{smallmatrix}1&0\\0&1\end{smallmatrix}\right]$ |
|  |  | $4v^4(1-v^2)^2$ | $\left[\begin{smallmatrix}1&1\\1&-1\end{smallmatrix}\right]$ |
| A666 | $1-3v^2$ | $4v^2(1-v^2)$ | $\left[\begin{smallmatrix}1&0&1\\0&1&1\end{smallmatrix}\right]$ |
| A3464 | $\left((1-v+v^2)^2-3v^2\right)$ $\times\left(1+v^2\right)^3$ | $-4v^6(1+v)^2(1-v)^6$ | $\left[\begin{smallmatrix}2&0&2\\0&2&2\end{smallmatrix}\right]$ |
|  |  | $8v^6(1+v)^2(1-v)^6$ | $\left[\begin{smallmatrix}1&2&1\\2&1&-1\end{smallmatrix}\right]$ |
|  |  | $8v^9(1-v)^2\times$ $\left(w^6-2w^4-w^3+2w^2+4w-8\right)$ | $\left[\begin{smallmatrix}1&0&1\\0&1&1\end{smallmatrix}\right]$ |
| A3636 | $(1-v+v^2)^2-3v^2$ | $4v^2(1+v^2)(1-v)^2$ | $\left[\begin{smallmatrix}1&0&1\\0&1&1\end{smallmatrix}\right]$ |
| A4444 | $1-2v-v^2$ | $4v(1-v^2)$ | $\left[\begin{smallmatrix}1&0\\0&1\end{smallmatrix}\right]$ |
| A33336 | $\left(1+3v^2\right)\left(3v^6-3v^4+12v^3-7v^2+4v-1\right)$ | $-4v^6(1-v^2)(1-v)^8$ | $\left[\begin{smallmatrix}2&0&2\\0&2&2\end{smallmatrix}\right]$ |
|  |  | $8v^5(1-v^4)(1-v)^6$ | $\left[\begin{smallmatrix}1&2&1\\2&1&-1\end{smallmatrix}\right]$ |
|  |  | $8v^3(1+v^2)(1-v)^3(2v^8+5v^7+v^6+6v^5+14v^4-7v^3+13v^2-4v+2)$ | $\left[\begin{smallmatrix}1&0&1\\0&1&1\end{smallmatrix}\right]$ |
| A33344 | $(1-3v)(1+v^2)$ | $-4v^2(v+1)(1-v)^3$ | $\left[\begin{smallmatrix}0\\2\end{smallmatrix}\right]$ |
|  |  | $4v^2(v+1)(1-v)^3$ | $\left[\begin{smallmatrix}1&1\\0&1\end{smallmatrix}\right]$ |
|  |  | $4v(1-v)(v^4+v^3+5v^2-v+2)$ | $\left[\begin{smallmatrix}0\\1\end{smallmatrix}\right]$ |
| A33434 | $(1-v^2)(1-2v-6v^3)$ $-v^4(9+7v^2)$ | $-4v^4(1+v)^2(1-v)^6$ | $\left[\begin{smallmatrix}2&0\\0&2\end{smallmatrix}\right]$ |
|  |  | $8v^3(1+v^2)(1+v)^2(1-v)^4$ | $\left[\begin{smallmatrix}1&1\\1&-1\end{smallmatrix}\right]$ |
|  |  | $8v^2(1-v^4)(1+3v^2)(1+v^2)(1-v)^2$ | $\left[\begin{smallmatrix}1&0\\0&1\end{smallmatrix}\right]$ |
| A333333 | $1-4v+v^2$ | $4v(1-v)^2$ | $\left[\begin{smallmatrix}1&0&1\\0&1&1\end{smallmatrix}\right]$ |

In App. D, we verify that $X_{\mathbf{k}}(v) \geq 0$ and give the solutions of $X_{\mathbf{k}}(v) = 0$ for $|v| \leq 1$. In addition to the single zero ($v_c > 0$) of the ferromagnetic model, bipartite lattices have a zero at $-v_c$ since $Z(-v) = Z(v)$, and the AF model has the same singularity as the F model. For A46C, A666, DA3464 and DA3636, the AF ground state and the lattice have the same periodicity (and the same unit cell), and $P_{\mathbf{k}}(-v) = P_{\mathbf{k}}(v)$, meaning that all polynomials $a_i$ are even. But for the other bipartite lattices, A488 and A4444, the unit cell of the AF ground state is twice as big as the unit cell of the lattice, and $P_{\mathbf{k}}(-v) = P_{\mathbf{k}+(\pi,\pi)}(v)$. The polynomial $a_0$ is not even.

A33434 and DA46C have a zero at some other negative $v$ (see Tab. 3 and discussion in Sec. 5.2). A3CC, A3464, A3636, A33366 and A333333 verifies $X_{\mathbf{k}_0}(-1) = 0$ for some $\mathbf{k}_0 \neq 0$, leading to a non-zero entropy of the AF model at $T = 0$ as shown in Sec. 5.2.

## 4 Energy and specific heat

Energy $e$ and specific heat $c_V$ per site are obtained from Eq. (19) (see App. E.1):

$$\frac{me}{J} = -n_l v - n_v(1-v) - \frac{1-v^2}{2} I_0(v), \tag{22}$$

$$\frac{mc_V}{\beta^2 J^2} = (1-v^2)(n_l - n_v - v I_0(v)) + \frac{(1-v^2)^2}{2} (I_1(v) - I_2(v)), \tag{23}$$

with

$$I_0(v) = \int_{BZ} \frac{d^2\mathbf{k}}{4\pi^2} \frac{X'_{\mathbf{k}}(v)}{X_{\mathbf{k}}(v)}, \tag{24a}$$

$$I_1(v) = \int_{BZ} \frac{d^2\mathbf{k}}{4\pi^2} \frac{X''_{\mathbf{k}}(v)}{X_{\mathbf{k}}(v)}, \tag{24b}$$

$$I_2(v) = \int_{BZ} \frac{d^2\mathbf{k}}{4\pi^2} \left[\frac{X'_{\mathbf{k}}(v)}{X_{\mathbf{k}}(v)}\right]^2, \tag{24c}$$

where the prime indicates a derivative with respect to $v$.

A singularity arises when $X_{\mathbf{k}}(v) = 0$ for some $k$ and $v$. For F models, it occurs when $\mathbf{k} = 0$ and $v = v_c$, the positive root of $a_0$ associated to a critical temperature $T_c$ that is known for all Archimedean lattices (see Tab. 1 for exact values and reference of first derivations), and Tab. 3 for numerical values). Fig. 5 shows the variation of $e$ and $c_V$ versus the temperature $T/J$ for Archimedean F-models. On this figure, we recover the strong correlation (quasi-linear dependancy) between the coordination number $z$ and the critical temperature $T_c$, which is not specific to the Ising model [34].

Near $v_c$, we define three similar dimensionless small parameters:

$$\epsilon = 1 - \frac{\beta}{\beta_c}, \qquad \epsilon_v = v - v_c, \qquad \epsilon_x = \beta_c J \epsilon = J(\beta_c - \beta). \tag{25}$$

As $X_{\mathbf{k}}(v) \sim \epsilon_v^2$, both integrals $I_0(v_c)$ and $I_2(v_c)$ are finite, while $I_1(v)$ exhibits a logarithmic divergence at $v_c$. In this section, we determine the lattice-dependent constants $A$ and $B$ such that near $v_c$:

$$c_V(\beta) + A \ln \left| 1 - \frac{\beta}{\beta_c} \right| \sim B. \tag{26}$$

In this section, the coefficient $A$ is obtained analytically and $B$ numerically for F models. Analytical expressions of $B$ are obtained for lattices with $n_f = 1$ (A3CC, A666, A3636, A444 and A333333).

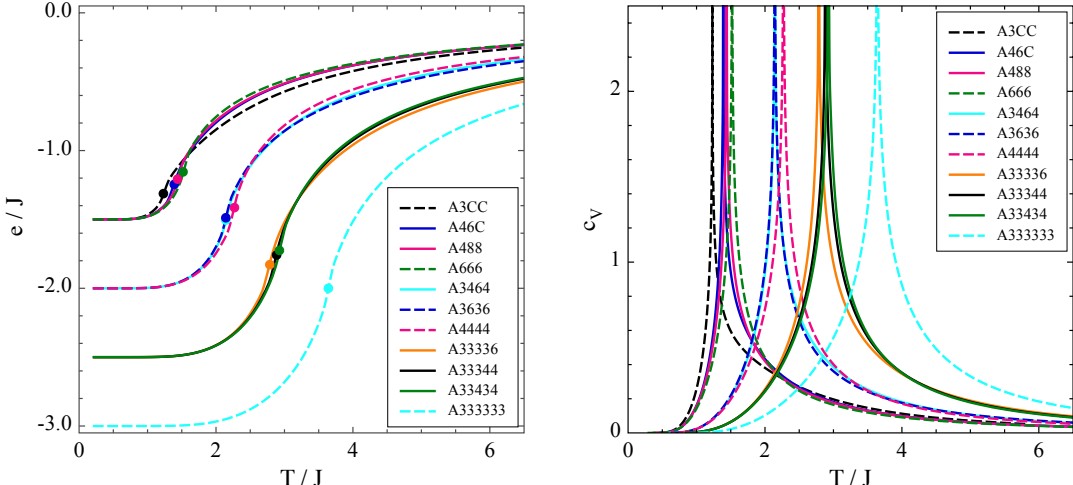

Figure 5: Energy (left) and specific heat (right) per spin versus temperature for Archimedean F models. Dashed lines are used for cases with $n_f = 1$ where analytical solutions are presented in App. E.4. On the left figure, dots stand for the critical temperature. We remark that all curves are grouped according to the coordination number $z$.

Fig. 6 shows the scaled energy $e$ and specific heat $c_V$ using these results. We see that the singular dominant term is accurate for $T/T_c \in [0.9, 1.05]$.

## 4.1 Evaluation of the dominant term $A \ln |1 - T_c/T|$ for ferromagnetic models

The divergent singularity in $c_V$ at $v_c$ comes from the integral $I_1(v)$, that determines $A$ in Eq. (1). Moreover, the singular behavior in $I_1(v)$ comes from the integration around $\mathbf{k} = 0$. For small $k = |\mathbf{k}|$:

$$X_{\mathbf{k}}(v) = a_0^2(v) + \sum_{i=1}^{n_f} a_i(v) \sum_{j=1}^{n_i} \frac{(\mathbf{c}_{ij} \cdot \mathbf{k})^2}{4} + O(k^4), \tag{27a}$$

$$X_{\mathbf{k}}''(v) = (a_0^2)''(v) + O(k^2), \tag{27b}$$

$$I_1(v) = \int_{BZ} \frac{d^2\mathbf{k}}{4\pi^2} \frac{(a_0^2)''(v)}{a_0^2(v) + \sum_{i=1}^{n_f} a_i(v) \sum_{j=1}^{n_i} \frac{(\mathbf{c}_{ij} \cdot \mathbf{k})^2}{4}} + O(1). \tag{28}$$

Then, near $v_c$:

$$I_1(v_c + \epsilon_v) \sim \frac{1}{4\pi^2} \int_0^{2\pi} d\theta \int_0^\rho \frac{2k\,dk}{\epsilon_v^2 + k^2 F(\theta)} \sim -\ln \epsilon_v^2 \int_0^{2\pi} \frac{d\theta}{F(\theta)} = -\frac{\ln |\epsilon_v|}{\pi \sqrt{\delta}}, \tag{29}$$

where

$$F(\theta) = \mu_{xx} \cos(\theta)^2 + \mu_{xy} \sin(2\theta) + \mu_{yy} \sin(\theta)^2, \tag{30a}$$

$$\delta = \mu_{xx} \mu_{yy} - \mu_{xy}^2, \tag{30b}$$

$$\mu_{ab} = \sum_{i=1}^{n_f} \frac{a_i(v_c)}{a_0'(v_c)^2} \sum_{j=1}^{n_i} \frac{c_{ij,a} c_{ij,b}}{4}. \tag{30c}$$

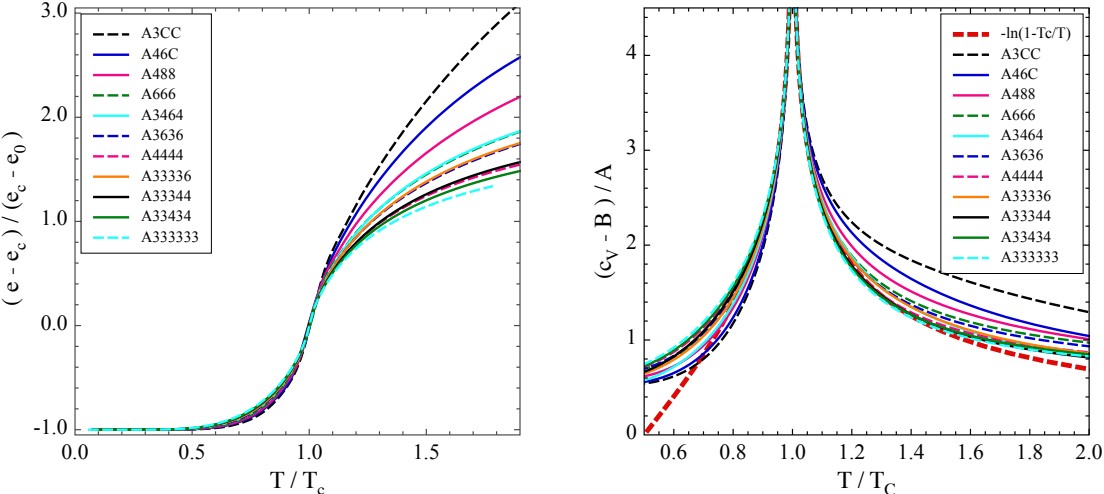

Figure 6: Same data as Fig. 5 for scaled energy (left) and specific heat (right) versus scaled temperature around the singularity. The energy at the transition and in the ground state are respectively denoted $e_c$ and $e_0$. We see that below $T_c$, scaled energies are very similar for all models. On the right figure, the red dashed line corresponds to the singular behavior: $-\ln|1 - T_c/T|$.

As $\epsilon_v$ is equivalent to $\epsilon$ up to a multiplicative constant for $v \to v_c$, we have from Eq. (23) one of the main result of this article, the analytical expression of $A$:

$$\frac{mA}{\beta_c^2 J^2} = \frac{(1 - v_c^2)^2}{2\pi\sqrt{\delta}}. \tag{31}$$

## 4.2 Evaluation of the subdominant term $B$ for ferromagnetic models

The evaluation of $B$ is more involved as it depends on $X_{\mathbf{k}}$ over the full BZ. We note that $\epsilon_v = (1 - v_c^2)\epsilon_x + o(\epsilon_x)$. Accordingly, we define $B_\epsilon$, $B_x$ and $B_v$ such that $c_V + A\ln\epsilon \sim B_\epsilon$, $c_V + A\ln\epsilon_x \sim B_x$ and $c_V + A\ln\epsilon_v \sim B_v$:

$$B \equiv B_\epsilon = B_x - A\ln(\beta_c J), \tag{32a}$$

$$B_x = B_v - A\ln(1 - v_c^2). \tag{32b}$$

After evaluating $I_0(v_c)$, $I_2(v_c)$ and

$$\mathcal{I}_1 = \lim_{\epsilon_v \to 0} \left( I_1(v_c + \epsilon_v) + \frac{1}{\pi\sqrt{\delta}} \ln|\epsilon_v| \right),$$

(see App. E for the calculations), we get the energy at the transition $e_c = e(v_c)$ and $B_v$:

$$\frac{me_c}{J} = -n_l v_c - n_v(1 - v_c) - \frac{1 - v_c^2}{2} I_0(v_c), \tag{33}$$

$$\frac{mB_v}{\beta_c^2 J^2} = (1 - v_c^2)(n_l - n_v - v_c I_0(v_c)) + \frac{(1 - v_c^2)^2}{2} (\mathcal{I}_1 - I_2(v_c)). \tag{34}$$

From this last equation, we get $B_x$ and $B = B_v - A\ln(\beta_c J(1 - v_c^2))$. Tab. 3 summaries the values of $T_c$, $A$, $B$, $B_x$, $B_v$ and of the energy $e_c$ at the transition. Despite strong variations in the parameters $z$, $m$, $n_l$, $n_v$ (see Tab. 1), the coefficients $A$ and $B$ of the singularity for the F models show no simple correlations with them. For completeness, the exact expression of $c_V$, $A$ and $B_x$ are given in Tab. 4 when $n_f = 1$ (derivation in App. E.4).

Table 3: Numerical values characterizing the F transition on all Archimedean lattices and their duals: critical temperature $T_c$ and its inverse $\beta_c$, energy $e_c$, coefficients $A$ and $B$ of the specific heat, see Eq. (26). Exact expressions are given for some of them in Tab. 4. As A4444=DA4444, A666=DA333333 and, DA666=A333333, results are identical for these couples of lattices. For the AF model, we have three cases: either the same singularity occurs (*) or a different one (#) or no singularity at all (no sign).

| lattice | $\beta_c J$ | $\frac{T_c}{J}$ | $\frac{e_c}{J}$ | $A$ | $B$ | $\frac{B_x}{\beta_c^2 J^2}$ | $\frac{B_v}{\beta_c^2 J^2}$ |
|---|---|---|---|---|---|---|---|
| A3CC | 0.81201 | 1.23151 | -1.31279 | 0.35600 | -0.18408 | -0.39161 | -0.71424 |
| A46C* | 0.71951 | 1.38983 | -1.24563 | 0.40405 | -0.20214 | -0.64739 | -1.02074 |
| A488* | 0.69507 | 1.43870 | -1.20731 | 0.43867 | -0.25016 | -0.84806 | -1.25539 |
| A666* | 0.65848 | 1.51865 | -1.15470 | 0.47811 | -0.30478 | -1.16363 | -1.61072 |
| A3464 | 0.46657 | 2.14332 | -1.50483 | 0.44790 | -0.21705 | -2.56570 | -2.99824 |
| A3636 | 0.46657 | 2.14332 | -1.48803 | 0.48006 | -0.29809 | -3.05062 | -3.51421 |
| A4444* | 0.44069 | 2.26919 | -1.41421 | 0.49454 | -0.30632 | -3.66394 | -4.14325 |
| A33336 | 0.35896 | 2.78584 | -1.82807 | 0.46346 | -0.24841 | -5.61307 | -6.06691 |
| A33344 | 0.34657 | 2.88539 | -1.75821 | 0.47792 | -0.61117 | -9.30453 | -9.77317 |
| A33434# | 0.79243 | 1.26194 | 1.27439 | 0.59740 | -0.70873 | -1.34998 | -1.89367 |
| A33434 | 0.34173 | 2.92626 | -1.72493 | 0.49484 | -0.30311 | -7.14531 | -7.63082 |
| A333333 | 0.27465 | 3.64096 | -2.00000 | 0.49907 | -0.30675 | -12.61595 | -13.10887 |
| DA3CC | 0.19972 | 5.00705 | -1.50419 | 0.25605 | -0.07836 | -12.30517 | -12.55954 |
| DA46C# | 0.57100 | 1.75131 | 0.86782 | 0.07804 | 0.061232 | 0.05368 | -0.02046 |
| DA46C | 0.24176 | 4.13629 | -1.72371 | 0.36119 | -0.13329 | -11.05429 | -11.41202 |
| DA488 | 0.25439 | 3.93101 | -1.84949 | 0.41677 | -0.20467 | -11.97887 | -12.39122 |
| DA666 | 0.27465 | 3.64096 | -2.00000 | 0.49907 | -0.30675 | -12.61595 | -13.10887 |
| DA3464* | 0.41572 | 2.40546 | -1.31874 | 0.41061 | -0.15893 | -3.00502 | -3.40433 |
| DA3636* | 0.41572 | 2.40546 | -1.33678 | 0.44009 | -0.24110 | -3.63019 | -4.05816 |
| DA4444 | 0.44069 | 2.26919 | -1.41421 | 0.49454 | -0.30632 | -3.66394 | -4.14325 |
| DA33336 | 0.53313 | 1.87572 | -1.16289 | 0.41593 | -0.17617 | -1.54027 | -1.93787 |
| DA33344 | 0.54931 | 1.82048 | -1.20423 | 0.45022 | -0.55619 | -2.73719 | -3.16643 |
| DA33434 | 0.55581 | 1.79917 | -1.22274 | 0.47522 | -0.28697 | -1.83239 | -2.28498 |
| DA333333* | 0.65848 | 1.51865 | -1.15470 | 0.47811 | -0.30478 | -1.16363 | -1.61072 |

## 5 Energy and entropy at $T = 0$

The ground state energy per site $e_0$ is obtained from Eq. (22) at $v_0 = J/|J|$, i.e. $v_0 = 1$ for F and $-1$ for AF. As the integral $I_0(v_0)$ is finite, we get:

$$e_0 = -\frac{|J|}{m} \left( n_l - n_v (1 - v_0) \right) . \tag{35}$$

The entropy per site $s_0$ at $T = 0$ is obtained using Eqs. (19), (20) and (22) from:

$$s_0 = \lim_{\beta \to \infty} (\beta e - \beta f) . \tag{36}$$

Table 4: Exact expression of quantities characterizing the F transition on the Archimedean with $n_f = 1$ and their dual lattices. The numerical evaluation is given for all of them in Tab. 3. $u_{2c}$ is defined in Eq. (E.32), $\mu = \sqrt{12 + 10\sqrt{3}}$ in column A3CC, and $\mu_D = \sqrt{36 + 30\sqrt{3}}$ in column DA3CC.

| lattice | A3CC | A666 | A4444 | A3636 | A333333 |
|---|---|---|---|---|---|
| $v_c$ | $\frac{\mu - 1 - \sqrt{3}}{4}$ | $1/\sqrt{3}$ | $\sqrt{2} - 1$ | $\frac{1 + \sqrt{3} - \sqrt[4]{12}}{2}$ | $2 - \sqrt{3}$ |
| $u_{2c}$ | $\frac{9\mu(17\sqrt{3} - 29)}{8} + \frac{144\sqrt{3} - 225}{4}$ | $6$ | $4$ | $18$ | $18$ |
| $e_c$ | $-\frac{2 + (13 - \sqrt{3})\mu}{48}$ | $-\frac{2}{\sqrt{3}}$ | $-\sqrt{2}$ | $-\frac{1}{3} - \frac{2}{\sqrt{3}}$ | $-2$ |
| $\frac{A}{\beta_c^2 J^2}$ | $\frac{u_{2c}}{3\sqrt{3}\pi}$ | $\frac{2\sqrt{3}}{\pi}$ | $\frac{8}{\pi}$ | $\frac{4\sqrt{3}}{\pi}$ | $\frac{12\sqrt{3}}{\pi}$ |
| $\frac{B_x}{\beta_c^2 J^2}$ | $\frac{(119 - 65\sqrt{3})\mu - 146\sqrt{3} + 216}{48} - \frac{A}{2\beta_c^2 J^2}\left(\ln\frac{u_{2c}}{18} + 2\right)$ | $\frac{A}{2}\ln\frac{3}{e^2} - \frac{2}{3}$ | $\frac{A}{2}\ln\frac{2}{e^2} - 2$ | $-A + \frac{2}{\sqrt{3}} - 2$ | $-A - 6$ |

| lattice | DA3CC | DA3636 |
|---|---|---|
| $v_c$ | $\frac{\mu_D - 3 - 3\sqrt{3}}{6}$ | $\frac{\sqrt{6\sqrt{3} - 9}}{3}$ |
| $u_{2c}$ | $\frac{3(5\sqrt{3} - 7)\mu_D + 162 - 60\sqrt{3}}{2}$ | $12\sqrt{3}$ |
| $e_c$ | $-\frac{(1 + 3\sqrt{3})\mu_D + 48 - 90\sqrt{3}}{36}$ | $-\frac{(5 + 3\sqrt{3})\sqrt{6\sqrt{3} - 9}}{9}$ |
| $\frac{A}{\beta_c^2 J^2}$ | $\frac{u_{2c}}{3\sqrt{3}\pi}$ | $\frac{8}{\pi}$ |
| $\frac{B_x}{\beta_c^2 J^2}$ | $\frac{(35 - 19\sqrt{3})\mu_D + 74\sqrt{3} - 192}{18} - \frac{A}{2\beta_c^2 J^2}\left(\ln\frac{u_{2c}}{18} + 2\right)$ | $-\frac{A}{4}\left(4 + \ln\frac{4}{3}\right) + \frac{10}{3} - \frac{22}{3\sqrt{3}}$ |

Around $T = 0$ we have $v - v_0 \sim -v_0 2 e^{-2\beta|J|}$, hence:

$$s_0 = \frac{1}{2m}\left[2(m + n_v - n_l)\ln 2 + \int_{BZ}\frac{d^2\mathbf{k}}{4\pi^2}\ln X_{\mathbf{k}}(\pm 1)\right]. \tag{37}$$

## 5.1 Ferromagnetic models ($J > 0$)

As expected, we recover from the previous formulae the ground state energy and the zero-entropy at $T = 0$:

$$e_0^{\mathrm{F}} = -\frac{J n_l}{m} = -\frac{J z}{2}, \qquad s_0^{\mathrm{F}} = 0. \tag{38}$$

Indeed, for Archimedean lattices we note that $a_{i \geq 1}(1) = 0$ and thus $X_{\mathbf{k}}(1) = a_0(1)^2$, cancelling $s_0^{\mathrm{F}}$ according to Eq. (18). For general lattices, as Eq. (18) always holds, in order to find a zero

Table 5: $T = 0$ energy and entropy of the Archimedean lattices for F ($J > 0$) and AF ($J < 0$) interactions.

| lattice | $\frac{e_0^{\mathrm{AF}}}{J}$ | $s_0^{\mathrm{AF}}$ | $\frac{e_0^{\mathrm{F}}}{J}$ | $s_0^{\mathrm{F}}$ |
|---|---|---|---|---|
| A3CC | 5/6 | 0.2509 | -3/2 | 0 |
| A46C | 3/2 | 0 | -3/2 | 0 |
| A488 | 3/2 | 0 | -3/2 | 0 |
| A666 | 3/2 | 0 | -3/2 | 0 |
| A3464 | 4/3 | 0.0538 | -2 | 0 |
| A3636 | 2/3 | 0.5018 | -2 | 0 |
| A4444 | 2 | 0 | -2 | 0 |
| A33336 | 7/6 | 0.0538 | -5/2 | 0 |
| A33344 | 3/2 | 0 | -5/2 | 0 |
| A33434 | 3/2 | 0 | -5/2 | 0 |
| A333333 | 1 | 0.3231 | -3 | 0 |

entropy, we obtain the following constraint:

$$\int_{BZ} \frac{d^2\mathbf{k}}{4\pi^2} \ln \frac{X_{\mathbf{k}}(1)}{a_0^2(1)} = 0. \tag{39}$$

A sufficient condition is $a_{i>0}(1) = 0$ ($a_i(\nu)$ divisible by $1-\nu$).

## 5.2 Antiferromagnetic models ($J < 0$) and application to Archimedean lattices

The ground state energy per site is:

$$e_0^{\mathrm{AF}} = -\frac{|J|}{m}(n_l - 2n_\nu). \tag{40}$$

We now evaluate the entropy at zero temperature $s_0^{\mathrm{AF}}$. Most non-bipartite lattices have $s_0^{\mathrm{AF}} \neq 0$. It has been calculated exactly for several lattices (triangular [35], kagome [36]), and numerically for others [37]. The non zero value finds its origin in Eq. (37), where $X_{\mathbf{k}}(-1)$ vanishes for some isolated $\mathbf{k}$-points. We present here the analytical expression and values of $s_0^{\mathrm{AF}}$ on all Archimedean lattices, summarized in Tab. 5 together with the numerical values of $e_0^{\mathrm{AF}}$.

**A46C, A488, A666, A4444, A33434:** We remark that $a_{i\geq 1}(-1) = 0$ and $|a_0(-1)| = a_0(1)$ for these lattices (see Tab. 2). The definition (16) and the property (18) lead to

$$s_0^{\mathrm{AF}} = 0.$$

This is expected for bipartite lattices, but may be more surprising for A33434 (usually called Shastry-Sutherland) as it possesses triangular faces. At minimal energy, only links shared by two triangles are frustrated. This is why the ground state degeneracy is only due to global spin flips (see Fig. 7a).

**A3CC, A3636:** $X_{\mathbf{k}}(-1) = 36 + 32\,\xi(\mathbf{k}) > 0$ with $\xi(\mathbf{k}) = \sin^2 \frac{k_x}{2} + \sin^2 \frac{k_y}{2} + \sin^2 \frac{k_x+k_y}{2}$, and $\int_{BZ} \frac{d^2\mathbf{k}}{4\pi^2} \ln(X_{\mathbf{k}}(-1)) \simeq 4.39729$ leading to

$$s_{0,\mathrm{A3636}}^{\mathrm{AF}} = 2 s_{0,\mathrm{A3CC}}^{\mathrm{AF}} \simeq 0.5018.$$

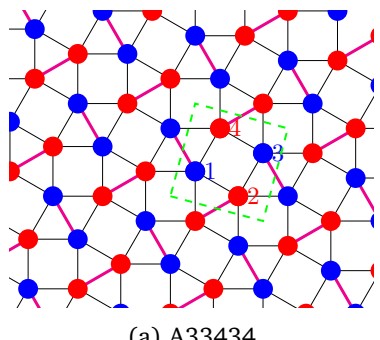
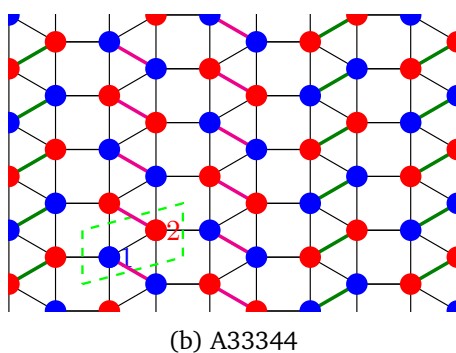

(a) A33434            (b) A33344

Figure 7: Ground states of the AF model on the A33434 and A33344 lattices. Blue and red points are for opposite spins. For A33434, magenta bonds are the only ones shared by two triangles. Frustrating them is the only way to minimize the energy: the ground state is completely fixed up to a global spin flip. For A33344, two choices are offered for each column of triangles: frustrating links oriented in the magenta or in the dark green direction, resulting in a sub-extensive entropy.

**A3464, A33336, A333333:** $X_{\mathbf{k}}(-1)=2^{\nu}\left(\frac{9}{4}-\xi(\mathbf{k})\right)$, with $\nu = 10,\ 12,\ 4$ respectively. Singularities arise at $\mathbf{k} \neq 0$: $X_{\pm\frac{2\pi}{3}}(-1) = 0$ and $\int_{BZ}\frac{d^2\mathbf{k}}{4\pi^2}\ln(X_{\mathbf{k}}(-1)) \simeq \nu\ln 2 - 0.74016$, leading to

$$s_{0,\text{A3464}}^{\text{AF}} = s_{0,\text{A33336}}^{\text{AF}} = s_{0,\text{A333333}}^{\text{AF}}/6 \simeq 0.0538\,.$$

**A33344:** $X_{\mathbf{k}}(-1) = 2^6\cos^2\frac{k_y}{2}$ and $\int_{BZ}\frac{d^2\mathbf{k}}{4\pi^2}\ln(X_{\mathbf{k}}(-1))=4\ln 2$, leading to

$$s_0^{\text{AF}} = 0\,.$$

Although the entropy per site of this lattice is zero at $T = 0$, the total entropy is sub-extensive, growing with the lattice linear size (see Fig. 7b).

Fig.8 shows the energy and specific heat for the AF Archimedean models. Bipartite lattices are not shown as they have the same variations as ferromagnetic models. Frustation (quantified by the entropy $s_0^{\text{AF}}$ at $T = 0$ and by $n_v$) has consequences on the shape of the specific heat, as $\ln 2 - s_0^{\text{AF}} = \int dT\, c_V(T)/T$. This effect is spectacular for the kagome lattice whose almost 3/4 of its entropy is conserved at $T = 0$.

# 6 Relations between different lattices

## 6.1 Duality for ferromagnetic models

The dual $\tilde{\mathcal{L}}$ of a planar graph or lattice $\mathcal{L}$ is obtained by replacing sites by faces and faces by sites. The link number is preserved, $n_l = \tilde{n}_l$, but they are rotated. An example of dual transformation is given for the kagome lattice A3636 (see Fig. 3).

In the F case ($\nu > 0$), it is known that the partition function $Z$ of $\mathcal{L}$ at high temperature is related to the partition function $\tilde{Z}$ of $\tilde{\mathcal{L}}$ at low temperature:

$$\tilde{Z}(\tilde{\nu}) = Z(\nu)2^{1-N_s}\left(\frac{\nu}{1-\nu^2}\right)^{-\frac{N_l}{2}}, \tag{41}$$

$$\tilde{\nu} = \frac{1-\nu}{1+\nu}\,. \tag{42}$$

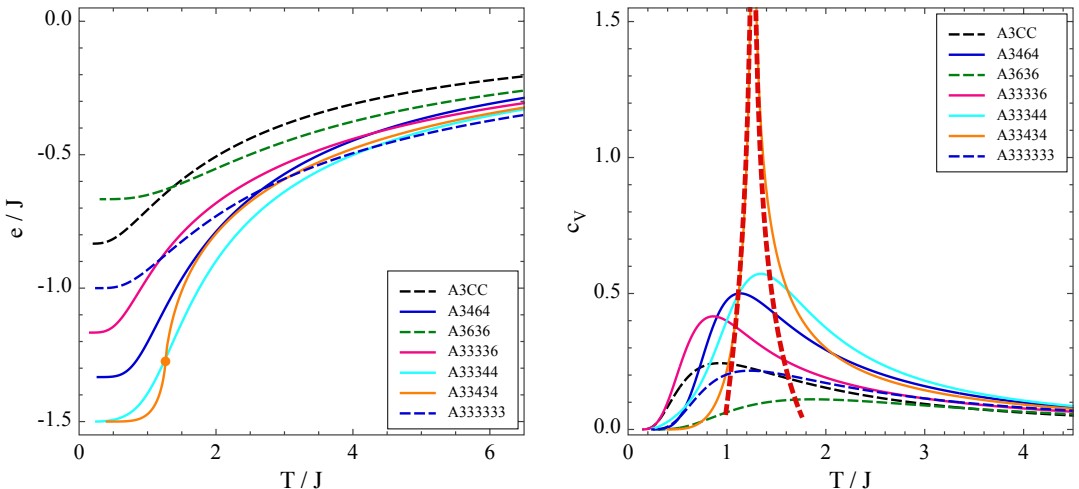

Figure 8: Same data as Fig. 5 for the antiferromagnetic models on non-bipartite Archimedean lattices. On the right figure, the red dashed line is $-A \ln|1 - \beta/\beta_c| + B$ for the A33344 lattice.

This was first discovered on the self-dual square lattice [19], then used on the regular triangular lattice and its dual the honeycomb lattice [35]. Appendix F provides two demonstrations for the relation between the Kac-Ward polynomials in $\mathcal{L}$ and $\tilde{\mathcal{L}}$ leading to the above formula and valid at any temperature (note that when $v$ goes from 0 to 1, $\tilde{v}$ goes from 1 to 0).

Eq. (41) allows to derive these relations between the quantities defined in Eqs. (16) and (17), on $\mathcal{L}$ and $\tilde{\mathcal{L}}$:

$$\frac{\tilde{a}_0(\tilde{v})(1+\tilde{v})^{\tilde{n}_v}}{a_0(v)(1+v)^{n_v}} = -\frac{2^{(n_l+m-\tilde{m})/2}}{(1+v)^{n_l}}, \tag{43}$$

$$\tilde{Y}_{\mathbf{k}}(\tilde{v}) = Y_{\mathbf{k}}(v), \tag{44}$$

where

$$Y_{\mathbf{k}}(v) = \frac{X_{\mathbf{k}}(v)}{a_0^2(v)} = 1 + \sum_{i=1}^{n_f} \frac{a_i(v)}{a_0^2(v)} \xi_i(\mathbf{k}). \tag{45}$$

In the thermodynamic limit, the relation between the free energies per site $f$ and $\tilde{f}$ is:

$$-\beta \tilde{m} \tilde{f}(\tilde{v}) = -\beta m f(v) - \frac{n_l}{2} \ln \frac{2v}{1-v^2} - \frac{m-\tilde{m}}{2} \ln 2, \tag{46}$$

or equivalently, in a symmetric way:

$$-\beta \tilde{m} \tilde{f}(\tilde{v}) - \frac{\tilde{m}}{2} \ln \frac{4\tilde{v}}{1-\tilde{v}^2} = -\beta m f(v) - \frac{m}{2} \ln \frac{4v}{1-v^2}. \tag{47}$$

The relations between the energies $e$ and $\tilde{e}$ and specific heats $c_V$ and $\tilde{c}_V$ per site of $\mathcal{L}$ and $\tilde{\mathcal{L}}$ are:

$$\frac{\tilde{m}\tilde{e}(\tilde{v})}{J} = \frac{-2v}{1-v^2} \frac{me(v)}{J} - n_l \frac{1+v^2}{1-v^2}, \tag{48}$$

$$\frac{\tilde{m}\tilde{c}_V(\tilde{v})}{\tilde{\beta}^2 J^2} = \frac{4v^2}{(1-v^2)^2} \left( \frac{mc_V(v)}{\beta^2 J^2} - \frac{1+v^2}{v} \frac{me(v)}{J} - 2n_l \right). \tag{49}$$

The symmetric version of Eqs. (48) and (49) (as in Eq. (47)) are given in App. G.

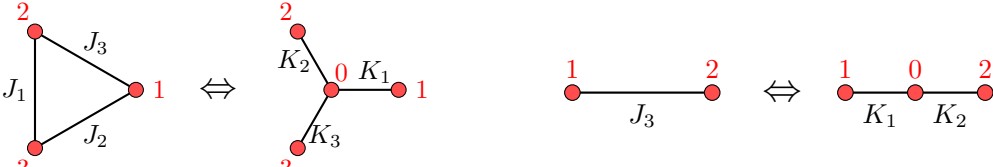

(a) Star-triangle transformation     (b) Decoration transformation

Figure 9: Star-triangle transformation in the general case (left) and for $K_3 = 0$ (right).

From Eq.(49), we deduce the relations between $\tilde{A}$ and $A$, and between $\tilde{B}$ and $B$:

$$\frac{\tilde{m}\tilde{A}}{\tilde{\beta}_c^2 J^2} = \frac{4v_c^2}{\left(1-v_c^2\right)^2}\frac{mA}{\beta_c^2 J^2}, \tag{50}$$

$$\frac{\tilde{m}\tilde{B}}{\tilde{\beta}_c^2 J^2} = \frac{4v_c^2}{\left(1-v_c^2\right)^2}\left(\frac{mB}{\beta_c^2 J^2} - \frac{1+v_c^2}{v_c}\frac{me_c}{J} - 2n_l + \frac{mA}{\beta_c^2 J^2}\ln\frac{(1-v_c^2)\beta_c}{2v_c\tilde{\beta}_c}\right). \tag{51}$$

## 6.2   Duality for antiferromagnetic models

Eqs. (43) and (44) for ferromagnetic models are still valid in the antiferromagnetic case, as they relate the two polynomials $P$ and $\tilde{P}$ independently of the values of $v$, while Eq. (42) would relate a $v$ corresponding to a positive temperature with a $\tilde{v} > 1$ (giving an unphysical negative temperature) and vice-versa. Then the antiferromagnetic thermodynamic functions have to be calculated separately on a lattice and its dual.

## 6.3   Star-triangle transformation

We have seen that the partition function of two dual lattices are related at different temperatures. We now give a well known relation between different lattices at the same temperature when they are related by a star-triangle transformation [33, 38], rederived in App. H.

  The Ising model on a lattice with a *triangle* and the one with a *star* obtained from the triangle by adding a spin in its center (see Fig. 9a) are equivalent (the ratio of their partition function does not depend on $\beta$) when

$$\tilde{t}_1 = \sqrt{\frac{(\tilde{s}_3\tilde{s}_1 + \tilde{s}_2)(\tilde{s}_1\tilde{s}_2 + \tilde{s}_3)}{(\tilde{s}_2\tilde{s}_3 + \tilde{s}_1)(\tilde{s}_1\tilde{s}_2\tilde{s}_3 + 1)}}, \tag{52}$$

and $\tilde{t}_2$ and $\tilde{t}_3$ verify Eq. (52) with a cyclic permutation of the indices, with

$$\tilde{t}_i = \exp(-2\beta J_i), \qquad \tilde{s}_i = \exp(-2\beta K_i). \tag{53}$$

  Equivalently, we have

$$s_1 = \sqrt{\frac{(t_3 t_1 + t_2)(t_1 t_2 + t_3)}{(t_2 t_3 + t_1)(t_1 t_2 t_3 + 1)}}, \tag{54}$$

and similarly for $s_2$ and $s_3$, where

$$t_i = \frac{1-\tilde{t}_i}{1+\tilde{t}_i} = \tanh\beta J_i, \tag{55a}$$

$$s_i = \frac{1-\tilde{s}_i}{1+\tilde{s}_i} = \tanh\beta K_i. \tag{55b}$$

Two special cases will be used below:

- All the exchanges are the same, $J_i = J$: then the site and link indices can be removed and

$$\tilde{t} = \sqrt{\frac{\tilde{s}}{\tilde{s}^2 - \tilde{s} + 1}} = \sqrt{\frac{1 - s^2}{1 + 3s^2}}, \tag{56a}$$

$$s = \sqrt{\frac{t}{t^2 - t + 1}} = \sqrt{\frac{1 - \tilde{t}^2}{1 + 3\tilde{t}^2}}. \tag{56b}$$

- $K_3 = 0$: then $\tilde{s}_3 = 1$ and $\tilde{t}_1 = \tilde{t}_2 = 1$, hence $J_1 = J_2 = 0$, which means that the remaining link $J_3$ is replaced by two links $K_1$ and $K_2$ (see Fig. 9b). We obtain:

$$\tilde{t}_3 = \frac{\tilde{s}_1 + \tilde{s}_2}{\tilde{s}_1 \tilde{s}_2 + 1}, \qquad t_3 = s_1 s_2. \tag{57}$$

## 6.4 Applications

Duality alone relates the critical $v_c$ in $\mathcal{L}$ to $\tilde{v}_c$ in $\tilde{\mathcal{L}}$. The values of the F critical temperatures $\tilde{T}_c$ of Laves lattice are obtained from $T_c$ and given in Tab. 3.

Using duality and star-triangle transformation allows to relate $v_c$ and other critical constants of several lattices with homogeneous coupling $J$:

- **A4444**: A4444 is self dual, hence $v_c = \tilde{v}_c$, i.e. $1 - 2v_c - v_c^2 = 0$ and $v_c = \sqrt{2} - 1$ [19]. We recover that $\tilde{A} = A$ and $\tilde{B} = B$.

- **A666, A333333**: A666 (critical $v_c$ denoted $v_6$) is the dual of A333333 (critical $v_c$ denoted $v_3$), hence $\tilde{v}_3 = v_6$. Triangle-star transformation turns back A333333 into A666 [33], hence using Eq. (56b) we obtain $v_6 = \sqrt{(1 - \tilde{v}_3^2)/(1 + 3\tilde{v}_3^2)}$, i.e. $v_6 = 1/\sqrt{3}$ and $v_3 = 2 - \sqrt{3}$. We verify that $e_c$, $v_v$, $A$ and $B$ of the triangular and honeycomb lattices respect the relations (48),(50) and (51).

- **A3636**: the kagome lattice (critical $v_c$ denoted $v_{36}$) through a star-triangle transformation becomes a variation of the honeycomb lattice (A666), in which every link is replaced with two links in a row [38]. Hence according to Eqs. (56b) and (57), we obtain:

$$v_6 = \sqrt{\left(\frac{1 - \tilde{v}_{36}^2}{1 + 3\tilde{v}_{36}^2}\right)^2} = \frac{1 - \tilde{v}_{36}^2}{1 + 3\tilde{v}_{36}^2}, \tag{58}$$

or $\tilde{v}_{36} = \sqrt{\frac{1 - v_6}{1 + 3v_6}} = \frac{\sqrt{3} - 1}{\sqrt[4]{12}}$ and $v_{36} = \frac{1 + \sqrt{3} - \sqrt[4]{12}}{2}$.

- **A3CC**: A3CC (critical $v_c$ denoted $v_C$) through a star-triangle transformation becomes a variation of the honeycomb lattice (A666), in which every link is replaced with three links in a row, with different energies [39, 40]. Hence Eqs. (56b) and (57) give:

$$\frac{1}{\sqrt{3}} = v_6 = v_C \sqrt{\left(\frac{v_C}{1 - v_C + v_C^2}\right)^2} = \frac{v_C^2}{1 - v_C + v_C^2}, \tag{59}$$

$$v_C = \frac{\sqrt{12 + 10\sqrt{3}} - 1 - \sqrt{3}}{4}. \tag{60}$$

More generally, duality and star-triangle transformations relate many lattices with several exchanges. For example, A488 with $J_2$ on links of squares and $J_1$ on links shared by two octogons is related to A33434 with $K_2$ on links of squares and $K_1$ on links shared by two triangles through a star-triangle transformation where a star $(J_1, J_2, J_2)$ turns into a triangle $(K_1/2, K_2, K_2)$ [41]. Similarly A46C with $J_1$ between a hexagon and a dodecagon, $J_2$ between a square and a dodecagon and $J_3$ between a square and a hexagon is related to A33336 with $K_2$ for the three sides of a triangle touching three hexagons, $K_3$ for a side of a hexagon and $K_1$ for the other links, through a star-triangle transformation, where a star $(J_1, J_2, J_3)$ turns into a triangle $(K_1/2, K_2, K_3)$. For both transformations, if all $J_i$s are equal, then $K_1$ is twice as big as other $K_i$s, as in [41].

# 7 Discussion and conclusion

In this article, we have reviewed the combinatorial method for calculating the free energy of Ising models on general planar lattices (for finite lattices or in the thermodynamic limit) and have provided the general formulae of the free energy, entropy, energy and specific heat. We also provide a review of the star-triangle and duality transformations, that relate different lattices.

All these formulae have been applied to the 11 Archimedean lattices and their duals for which we provide explicit expressions of the free energy (see Tab. 2) and deduce $T_c$, $A$ and $B$ (see Eq. (1) and Tab. 3). The value of $T_c$ was already known for these lattices, but the value of $A$ was only known for some of them, and the value of $B$ had not yet been evaluated. They have been calculated in this article, either analytically or numerically: when the unit cell contains many sites, the matrices to handle become large, and an analytical calculation is impossible after the last analytical step involving the calculation of a determinant (for which a code is provided in Supp. Mat. [25]). The zero temperature properties for F and AF models, and among them, the residual entropy $s_0^{AF}$ for extensively degenerated antiferromagnet (Tab. 5 for Archimedean lattices) have been determined. The exact $s_0^{AF}$ was known for the triangular [35] and kagome [36] lattice, but to our knowledge, only numerical evaluation were provided for A3CC, A3464 and A33336 [37]. Thus, we have either recorevered or calculated for the first time many quantities, usually spread in many papers but grouped here for a large set of commonly used lattices. We hope they will be useful to researchers to fit data with experimental or numerical results [42, 43].

The formulae in this article are directly applicable to any model on a planar lattice with a single type of link, for which the program given in Supp. Mat. [25] can be used. The formulae can be extended when several link types are present, as mentioned in Sec. 2.3 (examples of solutions in [13, 27]). However, extension to magnetization calculations, or to disordered systems [44] are left for future work.

## Acknowledgments

We thank Ethan Wanstock for the insightful work on the combinatorial method realized during his internship, prior to these calculations, and Jesper Jacobsen for giving us useful lectures and references. L. M. thanks Jeanne Colbois for interesting discussions on the Ising model.

# A   Proof of the Kac-Ward identity

We prove here Eq. (8), for a planar graph, in three steps. First, following Lis [21], we prove that the right-hand side $\sqrt{\det(I_{2N_l} - v\Lambda)}$ is a polynomial in $v$ (Sec. A.1), that we will denote $\sqrt{P}$. Then we define a polynomial $\bar{P}$ and prove it equal to $\sqrt{P}$ (Sec. A.2). Finally we prove $\bar{P}$ equal to the left-hand side of Eq. (8) when the graph is planar (Sec. A.3). Planarity is needed only at the end of Sec. A.3. Until there, the graph may be drawn on a flat torus.

## A.1   Proof that $\sqrt{\det(I_{2N_l} - v\Lambda)}$ is a polynomial denoted $\sqrt{P}$

We define:

$$P : \ v \mapsto \det(I_{2N_l} - v\Lambda), \tag{A.1}$$

$$\sqrt{P} : \ v \mapsto \sqrt{\det(I_{2N_l} - v\Lambda)}. \tag{A.2}$$

$P$ is a polynomial and is defined for any $v$, but $\sqrt{P}$ is a priori only defined for small $v$. However this function turns out to be a polynomial, which we will also denote $\sqrt{P}$ and use for all $v$. Then $\sqrt{P}(v)$ may be equal to $\pm\sqrt{P(v)}$ depending on $v$.

To prove that $\sqrt{P}$ is a polynomial, we prove that $\ln\det(I_{2N_l} - v\Lambda)$ is twice the logarithm of some other polynomial by considering the following series (that absolutely converges when $|v| < \frac{1}{2N_l}$):

$$\ln\det(I_{2N_l} - v\Lambda) = -\sum_{r=1}^{\infty} \frac{\mathrm{Tr}\,((v\Lambda)^r)}{r}, \tag{A.3}$$

(to obtain this formula, we have replaced $v\Lambda$ by a triangular matrix $A^{-1}v\Lambda A$, in which necessarily the diagonal coefficients are the eigenvalues of $v\Lambda$, then expanded the logarithms into series and grouped the terms in traces).

The diagonal coefficient $(\Lambda^r)_{l,l}$ is the sum of the weights of all closed paths of length $r$ on the lattice $\mathcal{L}$ departing from $l$ without backtracking, where the accumulation of direction changes (starting from the direction of $l$) determines the weight of each closed path $c$: $\pm 1$ depending on its turning number parity. We define $\Phi(c)$ as the opposite of the weight of $c$. For example, a non-self-intersecting closed path always has weight -1 if the graph is planar. The trace of $\Lambda^r$ is the sum of weights on all closed paths of length $r$.

To proceed with the proof, each link $l$ needs a different $v_l$ and $v$ is now a diagonal matrix in $v\Lambda$. An oriented link $l$ is denoted $-l$ when reversed and $\bar{l}$ when unoriented. Most of the time and eventually, we will choose $v_l = v_{-l} = v_{\bar{l}}$, but sometimes we will have $v_{-l} \neq v_l$ for some or all $l$. We define

$$\Phi_v(c) = \Phi(c) \prod_{l \in c} v_l. \tag{A.4}$$

The trace of $(v\Lambda)^r$ is the sum of $-\Phi_v(c)$ on all closed paths $c$ of length $r$.

But an oriented loop corresponds to several closed paths as we can choose several departures. Their number is the loop length, except when the loop is periodic, then it is the length of its smallest period. We extend the definitions of $\phi$ and $\phi_v$ to apply equally to oriented loops, since their values for a closed path does not depend on its starting point. Hence $\ln P$ is the sum of $\Phi_v(c)$ for all oriented loops $c$, but divided by $k$ if the loop is periodic with $k$ periods.

If each $l$ is given a different variable $v_l$ ($v_{-l} \neq v_l$ for all $l$) then $\det(I_{2N_l} - v\Lambda)$ is a polynomial of degree 1 in each of the $2N_l$ variables $v_l$. But a priori, $v_{-l} = v_l = v_{\bar{l}}$ for all $l$ and $P(v) = \det(I_{2N_l} - v\Lambda)$ is a polynomial of degree 2 in each of the $N_l$ variables $v_{\bar{l}}$. However, we denote $P(v_{l_0} = 0)$ when $v_{l_0} = 0$ and $v_l = v_{\bar{l}}$ for all other $l$. Hence $v_{-l_0} = v_{\bar{l_0}}$ and $P(v_{l_0} = 0)$ is a polynomial of degree 1 in $v_{\bar{l_0}}$ and 2 in any other $v_{\bar{l}}$. We choose an oriented link $l$. In

Eq. (A.3) the total contribution of closed paths containing both $l$ and $-l$ is 0, because in such a path we may reverse the subpath starting at the first occurence of $l$ or $-l$ and ending at the last occurence of the reverse link. This involution negates the weight. Furthermore the total contribution of loops containing $l$ and no $-l$ is equal to the contribution of loops containing $-l$ and no $l$, since reversing a loop does not change its weight. Hence removing contributions of loops containing $-l$ and halving contributions of loops containing no $l$ and no $-l$, halves the value of Eq. (A.3):

$$\frac{1}{2}\ln P(v) = \ln P(v_{-l}=0) - \frac{1}{2}\ln P(v_l=0, v_{-l}=0), \tag{A.5a}$$

$$\sqrt{P(v)} = P(v_{-l}=0)/\sqrt{P(v_l=0, v_{-l}=0)}, \tag{A.5b}$$

which is a polynomial of degree 1 in $v_{\bar{l}}$. This holds for any $l$. Hence $\sqrt{P(v)}$ is a polynomial of the $N_l$ variables $v_{\bar{l}}$ when they are small. We may denote this polynomial as $\sqrt{P}$. Then both polynomials $P$ and $\sqrt{P}^2$ have the same value for all small $v$, hence they are equal and have the same value for all $v$. Then we only have $\sqrt{P(v)} = |\sqrt{P}(v)|$, since $\sqrt{P}(v)$ may be far enough from 1 to be negative. Anyway polynomial $P$ is a square and $\sqrt{P}$ denotes the polynomial of constant term 1 and of square $P$.

## A.2 Definition of $\bar{P}$ and proof that $\sqrt{P} = \bar{P}$

Now that we are sure that both sides of Eq. (8) are polynomials, it remains to prove that they are equal, with as intermediate step to prove that $\sqrt{P}$ is equal to a polynomial $\bar{P}$ defined in this section, before showing that $\bar{P}$ is the left-hand side of Eq. (8) (next one).

An oriented loop $c$ is denoted $-c$ when reversed and $\bar{c}$ when unoriented. We can define $\Phi(\bar{c}) = \Phi(c)$ (defined in Sec. A.1), since $\Phi(-c) = \Phi(c) = \pm 1$. Similarly we can define $\Phi_v(\bar{c}) = \Phi_v(c)$, since $v_l = v_{-l}$ and $\Phi_v(-c) = \Phi_v(c)$.

Let $\vec{D}_\Lambda$ be the set of all sets $s$ of disjoint simple oriented (no duplicate oriented link) loops and $\bar{D}_\Lambda$ the same for unoriented loops. For $s \in \vec{D}_\Lambda$, we define $\Phi(s) = \prod_{c \in s} \Phi(c)$ and $\Phi_v(s) = \prod_{c \in s} \Phi_v(c)$, and similarly for $s \in \bar{D}_\Lambda$. We finally define

$$\bar{P} = \sum_{s \in \bar{D}_\Lambda} \Phi_v(s). \tag{A.6}$$

A preliminary step is to justify the following identity, first part of which is the Leibniz formula for determinants:

$$P = \sum_\sigma \epsilon_\sigma \prod_l (\delta_{l,\sigma(l)} - v_l \Lambda_{l,\sigma(l)}) = \sum_{s \in \vec{D}_\Lambda} \Phi_v(s). \tag{A.7}$$

A permutation $\sigma$ of all oriented links is the product of its cycles. Its signature $\epsilon_\sigma$ is the product of the signatures of its cycles. The product of factors $-v_l \Lambda_{l,\sigma(l)}$ corresponding to a cycle of $\sigma$ is not zero if and only if this cycle is an oriented loop $c$. Then this product is $-\Phi(c)\prod_{l \in c}(-v_l) = (-1)^{k+1}\Phi_v(c)$, where $(-1)^{k+1}$ is the signature of this cycle of length $k$.

A cycle $(l)$ of $\sigma$ of length 1, (i.e. $\sigma(l) = l$) may yield a factor $\delta_{l,l} = 1$, meaning this oriented link will not appear in this set of loops. It may also yield a factor $\Phi(c) = -v_l$ of $\Phi(s)$, if $c = (l)$ is a loop of length 1 of $s$. All of this proves Eq. (A.7).

To show that $P = \bar{P}^2$, we prove that *squarefree* (with no factor $v_{\bar{l}}^2$) monomials of both polynomials are the same. For any set $\lambda \subset \mathcal{L}$ of unoriented links the coefficients of $v^\lambda = \prod_{l \in \lambda} v_{\bar{l}}$ in $P$ and in $\bar{P}\bar{P}$ are equal, since any partition of $\lambda$ in a set $s$ of $r$ loops has the same contribution $2^r \Phi_v(s)$ in both polynomials: each loop may be in first factor $\bar{P}$ or in second factor $\bar{P}$, and it may be reversed or not in $P$. Since all the squarefree monomials of $\sqrt{P}^2$ and $\bar{P}^2$ are equal, and $\sqrt{P}$ and $\bar{P}$ are squarefree polynomials of constant term 1, we conclude that $\sqrt{P} = \bar{P}$.

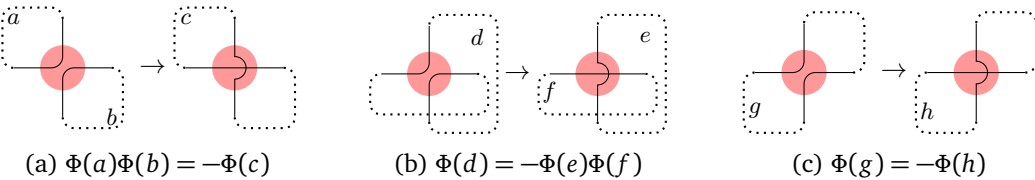

(a) $\Phi(a)\Phi(b) = -\Phi(c)$     (b) $\Phi(d) = -\Phi(e)\Phi(f)$     (c) $\Phi(g) = -\Phi(h)$

Figure 10: Operation of *loop crossing*. The red dot figures a site, and the function $\Phi$ acting on unoriented loops is defined in Sec. A.2.

## A.3   Proof that $\bar{P} = \sum v^r g_r$

To conclude, we now have to identify $\bar{P}$ with the left-hand side of Eq. (8). In the polynomial $\bar{P} = \sum_{\lambda \subset \mathcal{L}} \bar{p}_\lambda v^\lambda$, the coefficient $\bar{p}_\lambda$ is 0 if $\lambda$ is not even. We now prove that $\bar{p}_\lambda = \pm 1$ if $\lambda$ is even. Let $\lambda$ be an even set of unoriented links. Let $s$ be a partition of $\lambda$ into a set of loops. *Crossing* two parts of loops of $s$ (Fig. 10) changes the sign of $\Phi(s)$, either because we cross two parallel paths of oriented loops, changing the number of loops (see Figs. 10a and 10b), or because we reverse a part of a loop between two occurrences of a same site $a$, changing the evenness of its turning number (Fig. 10c). If in $\lambda$, the site $a$ has $2k$ adjacent links, then there are $\check{n}_k = 1 \times 3 \times \ldots (2k-1)$ ways to pair these $2k$ links. $(\check{n}_k + 1)/2$ of these pairings have an even number of crossings at site $a$, and $(\check{n}_k - 1)/2$ of them have an odd number of crossings. Hence the sum of the $\Phi(s)$ when $s$ assumes all of these $\check{n}_k$ pairings, equals a single $\Phi(s)$ with no crossing at $a$ in $s$. Hence $\bar{p}_\lambda$ is equal to $\Phi(s)$ for any partition $s$ of $\lambda$ in loops with no crossing at any site. This proves that $\bar{p}_\lambda = \pm 1$. So far $\mathcal{L}$ does not need to be planar.

If $\mathcal{L}$ is planar, $\bar{p}_\lambda = \Phi(s) = 1$ since any loop $c \in s$ is simple with no self-crossing and $\phi(c) = 1$. Hence $\bar{P}$ is the sum over even subgraphs $\lambda$ of $v^\lambda$, which directly gives the left-hand side of Eq. (8) when all $v_l$ are equal or the more general formula for link-dependent $v_{\bar{l}}$:

$$Z = 2^{N_s} \bar{P} / \prod_{\bar{l}} \sqrt{1 - v_{\bar{l}}^2}. \tag{A.8}$$

## B   The Kac Ward identity for graphs on a torus

When the graph is on a flat torus Eq. (8) turns into Eq. (12), recalled here:

$$\sum_{r=0}^{n_l} v^r g_r = \frac{1}{2} \left( \sqrt{P_{(0,\pi)}} + \sqrt{P_{(\pi,0)}} + \sqrt{P_{(\pi,\pi)}} - \sqrt{P_{\mathbf{0}}} \right)(v). \tag{B.1}$$

In this section, we prove this generalization of the Kac Ward identity on a torus in two steps, the first one being the analog of the proof in Sec. A for a planar graph. In a last subsection, we discuss the generalization to lattices drawn on surfaces of any genus $g$.

### B.1   Proof that $\bar{P}_\mathbf{k}^2 = P_\mathbf{k}$ when $2\mathbf{k} = 0$

We now try to replace $\Lambda$ by $\hat{W}(\mathbf{k})$ both in the proof that $\bar{P}^2 = P$ for a planar graph (Sec. A) and in the proof that $\bar{p}_\lambda = \Phi(s) = \pm 1$. Although the definitions of $\Phi(c)$ and $\Phi_v(c)$ change, the proofs still work provided any oriented loop still verifies $\Phi(c) = \Phi(-c) = \pm 1$, which holds when $2\mathbf{k} = 0$ as proven just below. This common value defines $\Phi(\bar{c})$ for the non oriented loop. Hence $\bar{P}$ is well defined.

*Proof.* Consider a single unit cell on a flat torus. For that graph, $\Lambda = \hat{W}(\mathbf{0})$ and the previous proofs work as long as planarity of graph is not needed. For any other $\mathbf{k}$, previous proofs need amendment. The upper side and lower side of a flat torus are identified. We may assume they are horizontal. For any link $l$ crossing this side upward (resp. downward), $v_l$ is replaced with $v_l e^{ik_y}$ (resp. $v_l e^{-ik_y}$) within Eq.(A.7). Similarly for every link $l$ crossing the non horizontal (vertical if the torus is square or rectangular but slanted in case of rhombus or rhomboid) side rightward (resp. leftward), $v_l$ is replaced with $v_l e^{ik_x}$ (resp. $v_l e^{-ik_x}$) within Eq.(A.7). Hence they turn $\Lambda = \hat{W}(\mathbf{0})$ into $\hat{W}(\mathbf{k})$ (see Eq. (10)), and they affect $\Phi(c)$ and $\Phi_v(c)$ for any oriented loop $c$: Eq. (A.4) still holds but $-\Phi(c)$ is now a product of coefficients of $\hat{W}(\mathbf{k})$. Now $\Phi(c)$ and $\Phi_v(c)$ implicitly depend on $\mathbf{k}$. To ensure $\Phi(c) = \Phi(-c)$ for any oriented loop $c$, we need $e^{ik_x} = e^{-ik_x}$ and $e^{ik_y} = e^{-ik_y}$, i.e. $k_x, k_y \in \{0, \pi\}$. Then $e^{ik_x} = \pm 1$, $e^{ik_y} = \pm 1$ and $\Phi(\bar{c}) = \Phi(-c) = \Phi(c) = \pm 1$. $\qquad\square$

## B.2 New expression of $\sum v^r g_r$

An even set of links, $\lambda$, may be split into a set of loops, $s$, with no crossing. Then $\bar{p}_\lambda = \Phi(s)$. If any loop in $s$ crosses each side of the torus an even number of times, then $\Phi(s) = 1$ whatever $\mathbf{k}$, and the contribution of $\lambda$ to Eq. (B.1) (meaning into coefficient of $v^\lambda$ in its right-hand side) is $(1 + 1 + 1 - 1)/2 = 1$. If an extra loop crosses once the horizontal side, the non horizontal side, or both of them, then $\Phi(s) = -1$ for half of the four values of $\mathbf{k}$, including $(0, 0)$, and the contribution will be $(-1 + 1 + 1 - (-1))/2 = 1$. This explains Eq. (B.1).

## B.3 Generalization to a graph on a surface of genus $g$

When the graph is drawn on a surface of genus $g$, this surface has $g$ handles and each handle is around a hole. A handle is a torus grafted on the surface. Handle $i$ is given $k_{x\,i}$. Hole $i$ is given $k_{y\,i}$. This allows to define $\bar{P}_{(k_{x\,1},k_{y\,1})\ldots(k_{x\,g},k_{y\,g})}$ as was done on the torus. Then Eq. (B.1) turns into

$$\sum_{r=0}^{n_l} v^r g_r = \frac{1}{2^g} \sum_{\substack{k_{x\,1}\ldots k_{x\,g}, \\ k_{y\,1}\ldots k_{y\,g} \in \{0,\pi\}}} \bar{P}_{(k_{x\,1},k_{y\,1})\ldots(k_{x\,g},k_{y\,g})} \prod_{i=1}^{g} (-1)^{\delta_{k_{x\,i}+k_{y\,i}}} . \tag{B.2}$$

Note that a link $l$ drawn on the surface is a curve making a fixed angle $\alpha_l$ with a vector field $\vec{e}_x$ on the surface. A priori we would like $\vec{e}_x \neq 0$ at each point of the surface. But then the surface would be a torus, because we cannot "comb" a connected closed orientable surface of genus $g \neq 1$. So we must allow a few points on the surface where $\vec{e}_x = 0$, but the turning number of field $\vec{e}_x$ around any such point must be even for the previous theory to work.

# C The Kac-Ward determinant properties

## C.1 Calculation of $P_{\mathbf{k}}$: Method and complexity

We prove here that $P_{\mathbf{k}}(v)$ defined in Eq. (11) can be stored efficiently in an array of integers and evaluate the complexity of Algo. 1 used in its calculation.

Each of the $n = 2n_l$ rows (or columns) of the matrix $\hat{W}(\mathbf{k})$ corresponds to an oriented link $l = l_i \to l_f$, where the site $l_i$ is chosen in a basic unit cell. $l_f$ may be in another cell. The coefficient $\hat{W}(\mathbf{k})_{l,l'}$ is not zero if $l'_i$ is the translate of $l_f$ by a Bravais lattice vector (translates of links may be successive steps of a path) and $l_f - l_i \neq l'_i - l'_f$ (no U-turn allowed). Its value is then given by Eqs. (9) and (10): $\hat{W}(\mathbf{k})_{l,l'} = e^{i[\alpha_{l'} - \alpha_l]/2} e^{i\mathbf{k}\cdot\mathbf{u}(l_f)}$, with $\alpha$ and $\mathbf{u}$ defined in Sec. 2.2. We will successively get rid of these two exponentials.

First, $P_{\mathbf{k}}$ does not change if we multiply row $l$ and divide column $l$ by $e^{i[\alpha_l]/2}$. The factor $e^{i[\alpha_{l'}-\alpha_l]/2}$ is replaced with $e^{i([\alpha_{l'}-\alpha_l]+[\alpha_l]-[\alpha_{l'}])/2}$ which is 1 when $|[\alpha_l]-[\alpha_{l'}]| < \pi$ and $-1$ otherwise.

The factor $e^{i\mathbf{k}\cdot\mathbf{u}(l_f)}$ is of the form $\phi^{x_{l_f}}\varphi^{y_{l_f}}$, where $\phi = e^{i\mathbf{k}\cdot\mathbf{e}_x}$, $\varphi = e^{i\mathbf{k}\cdot\mathbf{e}_y}$, $\mathbf{u}(l_f) = x_{l_f}\mathbf{e}_x + y_{l_f}\mathbf{e}_y$, where $x_{l_f}, y_{l_f} \in \mathbb{Z}$ and $(\mathbf{e}_x, \mathbf{e}_y)$ denotes a lattice basis. For all the lattices we will consider, we manage to have $x_{l_f}, y_{l_f} \in \{-1, 0, 1\}$. This way, the coefficients of the matrix $M$ used in Algo. 1 are Laurent polynomials in $\phi$ and $\varphi$, with exponents in $[-n, n]$ and $M$ can be stored as a four dimensional array $M[1..n][1..n][-n..n][-n..n]$ of integers.

Matrix multiplication $M *= \hat{W}(\mathbf{k})$ performs less than $n^2(2n+1)^2 z$ additions or subtractions of integers, since $\hat{W}(\mathbf{k})$ is a sparse matrix with on row $l$ only $c_{l_f}-1 \leq z-1$ non zero coefficients of the form $\pm\phi^{x_{l_f}}\varphi^{y_{l_f}}$, where $z$ is the greatest degree of all sites (and the coordination number for an Archimedean lattice). Therefore, the time of the Faddeev-Le Verrier algorithm is mainly spent for less than $n^3(2n+1)^2 z$ additions of integers, with $n = mz$ for a Archimedean lattice. However most of these additions are $0+0=0$, because the degree of a coefficient of $M$ after $j$ iterations is not $n$. It is not greater than $j$, and even not greater than $j/n_a$ for a diagonal coefficient (see C.3). Fortunately a straightforward transcription of Faddeev-Leverrier algorithm in Maple, avoids these useless additions and is very efficient if the coefficients of the matrix $\hat{W}(\mathbf{k})$ are all $0$, $\pm1$, $\pm\phi$, $\pm\varphi$, $\pm\phi*\varphi$, $\pm\phi/\varphi$, $\pm\varphi/\phi$, $\pm1/\phi/\varphi$, $\pm1/\phi$ or $\pm1/\varphi$:

```
with(LinearAlgebra):
n:=RowDimension(W):
M:=IdentityMatrix(n):
P:=1:
for j to n do
  M:=expand(M.W):
  p:=expand(-Trace(M)/j):
  P+=p*v^j:
  for k to n do
    M[k,k]:=expand(M[k,k]+p):
  od:
od:
```

Note the use of `expand` rather than `simplify`, which would be far much slower. This Maple program takes less than a second for any Archimedean lattice. It is provided in Supp. Mat. [25].

## C.2 Multiplicity of the factor $1+v$ in $P_{\mathbf{k}}$

In this section, we prove a property given in Sec. 2.5. Let $\mathrm{val}_{1+v}(Q(v))$ be the multiplicity of the factor $1+v$ in a polynomial $Q(v)$. Let $n_v$ be the minimal average number of links to remove per unit cell to make the graph bipartite. Then $\mathrm{val}_{1+v}(P_{\mathbf{k}}(v)) = 2n_v$.

*Proof.* $P_{\mathbf{k}}(v)$ is an integer Laurent polynomial in $v$, $\phi = e^{ik_x}$ and $\varphi = e^{ik_y}$. Let $a = \mathrm{val}_{1+v}(P_{\mathbf{k}}(v))$ be the multiplicity of the factor $1+v$ in this polynomial. Let $a_{\mathbf{k}}$ be the multiplicity of the factor $1+v$ in $P_k$ seen as a polynomial of $\mathbb{R}(v)$. It depends on $\mathbf{k}$, but $a_k \geq a$. However if $p$ and $q$ are two different prime numbers, polynomials $P_{\frac{2i\pi}{p}, \frac{2j\pi}{q}}$ for $i = 1, 2 \ldots p-1$ and $j = 1, 2 \ldots q-1$ are all isomorphic through automorphisms of field $\mathbb{Q}[\sqrt[pq]{1}]$. Hence $a_{\frac{2i\pi}{p}, \frac{2j\pi}{q}} = b_{p,q}$ does not depend on $i$ or $j$. We can prove that if $p$ and $q$ are large enough, then $b_{p,q} = a$. Otherwise the set $\{(\frac{2i\pi}{p}, \frac{2j\pi}{q}) \mid b_{p,q} > a, 0 < i < p, 0 < j < q\}$ would be dense in $[0, 2\pi]^2$ and for all $\mathbf{k}$ we would have $a_{\mathbf{k}} \geq a+1$ proving $\mathrm{val}_{1+v}(P_{\mathbf{k}}(v)) \geq a+1$ which is wrong.

Hence when $p$ and $q$ are large prime integers, in the product $\prod_{i=1}^{p}\prod_{j=1}^{q} P_{\frac{2i\pi}{p},\frac{2j\pi}{q}}$ the multiplicity of $1+\nu$ is $a$ for $(p-1)(q-1)$ factors and in $[a,n]$ for the $p+q-1$ other factors. Hence it is $\sim pqa$ for the whole product. Similarly, since $a_{\mathbf{k}} = a$ for allmost all $\mathbf{k}$, we may assume that in Eq. (13), the four square roots of products and their sum have the same behavior and have a factor $1+\nu$ of multiplicity equivalent to $\omega^2 a/2$. But for the sum, this multiplicity is the number of frustrated links, which is $\omega^2 n_\nu + O(\omega)$. Hence $n_\nu = a/2$. □

## C.3  Proof that $\nu^{n_a}$ divides $P_k - P_0$

Let $n_a$ be the length of the shortest loop with a non zero winding number in the single cell torus, $\nu^{n_a}$ divides $P_k - P_0$ (property given in Sec. 2.5).

*Proof.* As seen in Sec. A.1, the coefficient of degree $j$ in the Maclaurin series in $\nu$ of $\ln P_{\mathbf{k}}$ is the sum over all loops $c$ of length $j$, of $\Phi(c)$ or $\Phi(c)/i$ if $c$ is periodic with $i$ periods. If $j < n_a$ then both winding numbers of such a loop are 0, and $\Phi(c)$ does not depend on $\mathbf{k}$. Hence $\ln P_{\mathbf{k}} - \ln P_0 = O(\nu^{n_a})$ and $P_{\mathbf{k}} - P_0 = O(\nu^{n_a})$. □

## C.4  Proof that $\deg P_k \leq 2\sum_a \lfloor c_a/2 \rfloor$

In this section, we prove one of the properties given in Sec. 2.5, relating $\deg P_k$ and $c_a$, the degree of site $a$.

According to Eq. (A.7), $\deg P \leq n = \sum_a c_a$. If $s \in \vec{D}_\Lambda$ and loops of $s$ use all oriented links getting in (and out of) a site $a$, then if in $s$ we rewire connections in site $a$ and replace each and every $b \to a \to c$ by $c \to a \to b$, then $\Phi(s)$ and $\Phi_\nu(s)$ are multiplied by $(-1)^{c_a}$. Then these terms cancel in $\sum_{s \in \vec{D}_\Lambda} \Phi_\nu(s)$ if $c_a$ is odd. Hence contribution to $\deg P$ of links getting out of site $a$ is at most $c_a$ if even, or else $c_a - 1$. Hence $\deg P \leq 2\sum_a \lfloor c_a/2 \rfloor$. This proof still works with $P_k$ instead of $P$.

## C.5  Value of $\bar{P}_k(1)$ for $2k = 0$

In this section, we prove one of the properties given in Sec. 2.5 about special values of $\bar{P}_{\mathbf{k}}(1)$ for $2\mathbf{k} = 0$. The symmetric difference $\lambda \triangle \lambda' = (\lambda \cup \lambda') \setminus (\lambda \cap \lambda') = (\lambda \setminus \lambda') \cup (\lambda' \setminus \lambda)$ of two even subgraphs of $\mathcal{L}$ is even. Hence the set $E_\mathcal{L}$ of all even subgraphs of $\mathcal{L}$ is a vector space over field $F_2 = \mathbb{Z}/2\mathbb{Z}$. Set $E_\mathcal{L}$ has a partition in four parts $E_{00}$, $E_{01}$, $E_{10}$ and $E_{11}$, where $\lambda \in E_{\epsilon_1 \epsilon_2}$ if $\epsilon_1$ (respectively $\epsilon_2$) is congruent modulo 2 to the number of links of $\lambda$ which cross the horizontal (resp. non horizontal) side of torus. These four parts have the same cardinality since $\lambda \mapsto \lambda \triangle \lambda_0$ is a bijection from $E_{00}$ onto $E_{\epsilon_1 \epsilon_2}$ if $\lambda_0 \in E_{\epsilon_1 \epsilon_2}$. The set $E_{00}$ is a vector space of dimension $\tilde{m} - 1$, since (perimeters of) faces but one form a basis. Hence $\#E_{00} = \#E_{01} = \#E_{10} = \#E_{11} = 2^{\tilde{m}-1}$ and

$$\bar{P}_0(1) = \#E_{00} - \#E_{01} - \#E_{10} - \#E_{11} = -2^{\tilde{m}}, \tag{C.1}$$

$$\bar{P}_{0\pi}(1) = \#E_{00} - \#E_{01} + \#E_{10} + \#E_{11} = 2^{\tilde{m}}, \tag{C.2}$$

$$\bar{P}_{\pi 0}(1) = \#E_{00} + \#E_{01} - \#E_{10} + \#E_{11} = 2^{\tilde{m}}, \tag{C.3}$$

$$\bar{P}_{\pi\pi}(1) = \#E_{00} + \#E_{01} + \#E_{10} - \#E_{11} = 2^{\tilde{m}}, \tag{C.4}$$

$$(\bar{P}_{\pi\pi}(1) + \bar{P}_{\pi 0}(1) + \bar{P}_{0\pi}(1) - \bar{P}_0(1))/2 = \#E_{00} + \#E_{01} + \#E_{10} + \#E_{11} = \#E_\mathcal{L} = 2^{\tilde{m}+1}. \tag{C.5}$$

Furthermore $\bar{P}_{\pi\pi}(0) = \bar{P}_{\pi 0}(0) = \bar{P}_{0\pi}(0) = \bar{P}_0(0) = 1$. Since $\bar{P}_0(0) > 0$ and $\bar{P}_0(1) < 0$, there exists $\nu_c \in (0,1)$ such that $\bar{P}_0(\nu_c) = 0$ implying a finite temperature phase transition in ferromagnetic 2d models.

All of this holds when the graph is connected and double-periodic. If ever it is only simple-periodic, like the unbalanced ladder of Fig. 4, then (for instance)

$\#E_{00} = \#E_{01} > \#E_{10} = \#E_{11} = 0$ and $\bar{P}_{\mathbf{0}}(1) = 0$ (ordering only occurs at $T = 0$ in 1d ferromagnetic models).

# D  Signs and zeroes of characteristic polynomials for Archimedean lattices

In this appendix, we show that $X_{\mathbf{k}}(v) \geq 0$ (see Eq. (16)) and determine the zeroes for Archimedean lattices for the F ($v > 0$) and AF Ising model ($v < 0$). Value $v = 0$ corresponds to $\beta = 0$ (infinite temperature), whereas $v = \pm 1$ corresponds to $T = 0$ (+1 for F, −1 for AF).

**Bipartite lattices A46C, A488, A666 and A4444:**  As these lattices are not frustrated, the energy depends only on $|v|$, as it will be proved from the expression of $X_{\mathbf{k}}$ in this paragraph. A666 and A46C have a magnetic unit cell for the AF order (see Fig. 11) which is the lattice unit cell. It leads to even polynomials $a_i(v)$ and the proof is obvious. But for A488 and A4444, the magnetic unit cell is doubled in the AF case (see Fig. 11). The equivalence between the F and AF models appears in $X_{\mathbf{k}}$: the sum $a_0^2(v) + a_1(v)\xi_1(\mathbf{k})$ is unchanged when replacing $v$ with $-v$ and $\mathbf{k}$ with $\mathbf{k}+(\pi,\pi)$ since $2a_1(v) = a_0^2(-v) - a_0^2(v)$ and $\xi_1(\mathbf{k}) = 2 - \xi_1(\mathbf{k}+(\pi,\pi))$. A4444 has no other terms, while for A488, the extra term $a_2(v)$ is even.

Hence, for these 4 lattices, it is sufficient to search the zeros and sign of $X_{\mathbf{k}}(v)$ for the F case (see next paragraphs).

**Lattices A488 and A4444:**  $\xi_1(\mathbf{k})$ is between 0 ($\mathbf{k} = 0$) and 2 ($\mathbf{k} = (\pi,\pi)$) and from previous paragraph we know that $2a_1(v) = a_0^2(-v) - a_0^2(v)$. Hence $a_0^2(v) + a_1(v)\xi_1(\mathbf{k})$ is between $a_0^2(v)$ and $a_0^2(-v)$ for any $\mathbf{k}$. For A488, the extra term $a_2(v)\xi_2(\mathbf{k}) \geq 0$ since $a_2(v)$ is a square. Thus for both lattices $X_{\mathbf{k}}(v) \geq 0$. Furthermore $a_0(v)$ vanishes only at $v_c$ (given in Tab. 2) and the two zeroes of $X_{\mathbf{k}}(v)$ are for $\{v=v_c, \mathbf{k}=0\}$ and $\{v=-v_c, \mathbf{k}=(\pi,\pi)\}$.

**Lattices A3CC and A3636:**  $a_0(v)$ vanishes only at $v_c>0$ (given in Tab. 2), $a_1(v)\geq 0$. Thus $X_{\mathbf{k}}(v)\geq 0$ and the single zero of $X_{\mathbf{k}}(v)$ is for $\{v=v_c, \mathbf{k}=0\}$.

**Lattices A46C, A3464 and A33336** have a rotational symmetry of order 6 and share the same three $\xi_i(\mathbf{k}) \in [0, \frac{9}{4}]$ related by:

$$\xi_1(\mathbf{k}) + 2\xi_2(\mathbf{k}) = 10\xi_3(\mathbf{k}) - 4\xi_3^2(\mathbf{k}). \tag{D.1}$$

Eliminating $\xi_1(\mathbf{k})$ in $X_{\mathbf{k}}(v)$ with Eq. (D.1) gives:

$$X_{\mathbf{k}}(v) = a_0^2(v) + (a_2(v) - 2a_1(v))\xi_2(\mathbf{k}) - 4a_1(v)\xi_3^2(\mathbf{k}) + (a_3(v) + 10a_1(v))\xi_3(\mathbf{k}). \tag{D.2}$$

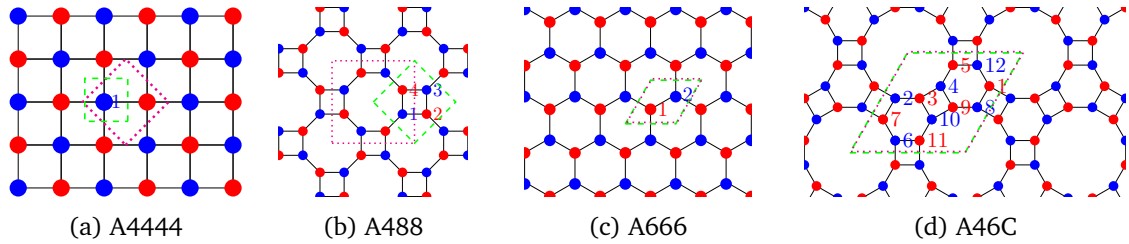

(a) A4444          (b) A488          (c) A666          (d) A46C

Figure 11: AF ground state on bipartite Archimedean lattices. The green and magenta cells are respectively the lattice and magnetic order unit cells.

For $v > 0$, $a_0(v)$ vanishes at $v_c$ (given in Tab. 2). Furthermore $a_1(v) \leq 0$, $a_2(v) \geq 0$ and $a_3(v) + 10a_1(v) \geq 0$, hence $X_{\mathbf{k}}(v) \geq a_0^2(v)$. Thus $X_{\mathbf{k}}(v)$ vanishes for $\{v = v_c, \mathbf{k} = 0\}$ and is positive elsewhere. This holds for all three lattices.

The case $v < 0$ is handled differently for each lattice.

**A46C:** Polynomials $a_i(v)$ are even and $X_{\mathbf{k}}(v) = X_{\mathbf{k}}(-v)$. Thus $X_{\mathbf{k}}(v)$ vanishes for $\{v = \pm v_c, \mathbf{k} = 0\}$ and is positive elsewhere. The singularities at $v_c$ and $-v_c$ are identical.

**A3464:** For negative $v$, eliminating $\xi_2(\mathbf{k})$ in $X_{\mathbf{k}}(v)$ with Eq. (D.1) gives:

$$X_{\mathbf{k}}(v) = a_0^2(v) + (a_1(v) - a_2(v)/2)\xi_1(\mathbf{k}) - 2a_2(v)\xi_3(\mathbf{k})^2 + (a_3(v) + 5a_2(v))\xi_3(\mathbf{k}). \quad \text{(D.3)}$$

Here all terms are negative but $a_0^2(v)$. Setting $\xi_1(\mathbf{k})$ and $\xi_3(\mathbf{k})$ to their maxima $9/4$ gives a lower bound $X_{\mathbf{k}}(v) \geq a_0^2(v) + \frac{9}{4}(a_1(v) + a_3(v))$ which is positive and vanishes at $v = -1$. $X_{\mathbf{k}}(v = -1)$ vanishes for $\mathbf{k} = \pm(\frac{2\pi}{3}, \frac{2\pi}{3})$.

**A33336:** For $v < 0$, all $a_i(v)$ are negative. $X_{\mathbf{k}}(v)$ vanishes only at $v = -1$, for $\xi_3(\mathbf{k}) = \frac{9}{4}$ that is $\mathbf{k} = \pm(\frac{2\pi}{3}, \frac{2\pi}{3})$. To prove this we will search for minima of $X_{\mathbf{k}}(v)$. With $\phi = e^{ik_x}$ and $\varphi = e^{ik_y}$ we notice that

$$\frac{\partial X_{\mathbf{k}}(v)}{\partial \phi} \frac{4\phi^3\varphi^2}{1 - \phi^2\varphi} \quad \text{(D.4)}$$
$$= 2a_1(v)(\phi^2\varphi + 1)(1 + \varphi^2) + a_2(v)(\phi\varphi^3 + \phi + 2\phi^2\varphi^2/2\varphi) + a_3(v)\phi\varphi(1 + \varphi),$$

$$\left(\frac{\partial X_{\mathbf{k}}(v)}{\partial \phi} \frac{\phi + \phi^2}{1 - \phi^2\varphi} - \frac{\partial X_{\mathbf{k}}(v)}{\partial \varphi} \frac{\varphi + \varphi^2}{1 - \phi\varphi^2}\right) 4\phi^2\varphi^2 \quad \text{(D.5)}$$
$$= (2a_1(v) - a_2(v))(\phi - \varphi)(1 - \phi)(1 - \varphi)(1 - \phi\varphi).$$

At the minimum of $X_{\mathbf{k}}(v)$ for a fixed $v$, $\partial X_{\mathbf{k}}(v)/\partial \phi = \partial X_{\mathbf{k}}(v)/\partial \varphi = 0$. Hence in Eq. (D.5) either $1 - \phi^2\varphi = 0$ or $1 - \phi\varphi^2 = 0$ or the right-hand side is 0. At least one of the following seven equations holds:

$$\phi\varphi^2 = 1, \qquad \phi = \varphi, \qquad \phi^2\varphi = 1, \quad \text{(D.6)}$$
$$\phi = 1, \qquad \varphi = 1, \qquad \phi\varphi = 1, \quad \text{(D.7)}$$
$$2a_1(v) = a_2(v). \quad \text{(D.8)}$$

- Eq. (D.8) and (D.2) give $X_{\mathbf{k}}(v) = a_0^2(v) + a_3(v)\xi_3(\mathbf{k}) + a_1(v)(10\xi_3(\mathbf{k}) - 4\xi_3^2(\mathbf{k}))$. It is a monotonic function of $\xi_3(\mathbf{k})$ which is extremal like $\xi_3(\mathbf{k})$ when the three equations (D.6) hold. Hence we may ignore Eq. (D.8).

- If 2 of the 3 equations (D.6) hold, then they all hold and either $\mathbf{k} = 0$ and $X_{\mathbf{k}}(v) = a_0^2(v) > 0$, or $\mathbf{k}_0 = \pm\left(\frac{2\pi}{3}, \frac{2\pi}{3}\right)$ and $X_{\mathbf{k}}(v) = a_0^2(v) + \frac{9}{4}a_1(v) + \frac{9}{4}a_3(v) \geq 0$. It vanishes only at $v = -1$. These three $\mathbf{k}$-points are indeed the extrema of $X_{\mathbf{k}}(v)$.

- Similarly if two of the three equations (D.7) hold, then they all hold and $\mathbf{k} = 0$ and $X_{\mathbf{k}}(v) = a_0^2(v) > 0$.

- When one equation of (D.6) and one equation of (D.7) hold, like $\phi = 1 = \phi\varphi^2$, there are only three new cases: $\mathbf{k} = (\pi, 0)$, $\mathbf{k} = (0, \pi)$ and $\mathbf{k} = (\pi, \pi)$. Then $X_{\mathbf{k}}(v)$ is the square of polynomial $\bar{P}_{\mathbf{k}}(v) = 7v^8 + 6v^6 + 12v^5 + 8v^4 - 8v^3 + 10v^2 - 4v + 1 > 0$.

- From now on we will assume that only one of the six equations (D.6) and (D.7) holds. But $(\phi, \varphi) \mapsto (\varphi, 1/\phi\varphi)$ does not change $X_{\mathbf{k}}(v)$ and permutes the three equations (D.6) (or (D.7)). Hence we may also assume $\phi\varphi^2 \neq 1 \neq \phi^2\varphi$ and $\phi \neq 1 \neq \phi\varphi$. It remains two cases to study: $\varphi = 1$ or $\phi = \varphi$, combined with a null right-hand side of Eq. (D.4).

- $\varphi = 1$ and $0 = 2a_1(v)(1 + \phi^2) + a_2(v)(1 + \phi + \phi^2) + a_3(v)\phi$. Thus $2\cos k_x = \phi + \frac{1}{\phi} = -\frac{a_2(v)+a_3(v)}{a_2(v)+2a_1(v)} < -3$. This is impossible.

- $\phi = \varphi$ and Eq. (D.4) give

$$0=(a_3(v)\phi^2+2a_1(v)(1+\phi^2)(1-\phi+\phi^2)+3a_2(v)\phi(1-\phi+\phi^2))(1+\phi).$$

But $\phi \neq -1$. Hence $x = \phi + 1/\phi$ is a root of equation

$$2a_1(v)x(x-1)+3a_2(v)(x-1)+a_3(v)=0.$$

This polynomial in $x$ has a negative discriminant if $-1<v<0$ and is equal to -4096 at $v = -1$. Hence this equation has no root.

This proves that $X_{\mathbf{k}}(v)$ is minimal at $\mathbf{k} = \pm\left(\frac{2\pi}{3}, \frac{2\pi}{3}\right)$ and maximal at $\mathbf{k} = \mathbf{0}$. It never vanishes.

**A666:** $a_0(v)$ and $a_1(v)$ are even function of $v$. $a_0(v)$ vanishes at $\pm v_c = \pm 1/\sqrt{3}$ and $a_1(v) \geq 0$. Thus $X_{\mathbf{k}}(v) \geq 0$ and $X_{\mathbf{k}}(v)$ vanishes at $\{v=\pm v_c, \mathbf{k}=0\}$ and the singularity occurs at $\pm v_c$.

**A33344:** $a_0(v)$ vanishes only at $v=v_c$ (given in Tab. 2). Let $t = \cos(k_x + \frac{k_y}{2})$ and $z = \cos\frac{k_y}{2}$. Then $\xi_2(\mathbf{k}) = 1 - zt$, $\xi_3(\mathbf{k}) = 1 - z^2$ and $\xi_1(\mathbf{k}) = 4z^2 - 4z^4$. Then $X_{\mathbf{k}}(v)$ becomes:

$$X(v,z,t) = a_0^2(v) + a_2(v)(1-zt-4z^2+4z^4) + a_3(v)(1-z^2), \tag{D.9}$$

$X_{\mathbf{k}}(v,z,t)$ is always between $X_{\mathbf{k}}(v,z,1)$ and $X_{\mathbf{k}}(v,z,-1) = X_{\mathbf{k}}(v,-z,1)$. Then $X_{\mathbf{k}}(v,z,t)$ is positive for all $t,z$ in $[-1,1]$ iff $X_{\mathbf{k}}(v,z,1)$ is positive for all $z$. It is true for negative $v$ since $a_2(v)z^4 \geq 0$ and $X_{\mathbf{k}}(v,z,t) - 4a_2(v)z^4$ is a polynomial of degree 2 in $z$, of discriminant $4a_2^2(v) + 4(a_0^2(v) + a_2(v) + a_3(v))(4a_2(v) + a_3(v)) \leq 0$ and $4a_2(v) + a_3(v) \leq 0$. Thus $X_{\mathbf{k}}(v) \geq 0$ for $v < 0$, and vanishes for $v = -1$ ($a_1(-1) = a_2(-1) = 0$, $a_0^2(-1) = -a_3(-1) = 64$) and $z = 0$ that is for $\mathbf{k} = (k_x, \pi)$ (see 5.2).

For positive $v$, $10a_2(v) \leq a_3(v)$ and $(1-zt-4z^2+4z^4)/10 + (1-z^2) \geq 0$ and vanishes for $z = t = \pm 1$ ($\mathbf{k} = 0$). Then for positive $v$, $X_{\mathbf{k}}(v) \geq 0$ and vanishes only at $\{v = 1/3; \mathbf{k} = 0\}$.

**A33434:** $a_0(v)$ vanishes at $v=v_c>0$ (given in Tab. 2) but also at $v_c^{AF} \sim -0.659784$. For $v>0$, we have $a_1(v) \leq 0$, and $a_2(v) > 0$ and $a_3(v) > 0$. The relation between the $\xi_i(\mathbf{k})$ is:

$$4\xi_3(\mathbf{k})^2 - 8\xi_3(\mathbf{k}) + \xi_1(\mathbf{k}) + 2\xi_2(\mathbf{k}) = 0. \tag{D.10}$$

Substituting $\xi_1(\mathbf{k})$ in $X_{\mathbf{k}}(v)$ gives:

$$X_{\mathbf{k}}(v)=(-2a_1(v) + a_2(v))\xi_2(\mathbf{k}) - 4a_1(v)\xi_3(\mathbf{k})^2 + (8a_1(v) + a_3(v))\xi_3(\mathbf{k}) + a_0^2(v). \tag{D.11}$$

For positive $v$, as $8a_1(v) + a_3(v) \geq 0$, $X_{\mathbf{k}}(v) \geq 0$ and vanishes only at $\{v=v_c; \mathbf{k}=0\}$. For negative $v$, $a_1(v) \leq 0$, $a_2(v) \leq 0$ and $a_3(v) \geq 0$, with $a_1(v)/a_3(v) > -0.1$ and $a_2(v)/a_3(v) > -0.2$, thus we have $X \geq a_0^2(v) + a_3(v)(-\xi_1(\mathbf{k})/10 - \xi_2(\mathbf{k})/5 + \xi_3(\mathbf{k}))$. Substituting $\xi_1(\mathbf{k})$ in $X_{\mathbf{k}}(v)$ gives $a_0^2(v) + (2\xi_3(\mathbf{k})^2 + \xi_3(\mathbf{k}))a_3(v)/5$ which is positive and vanishes for $\{v=v_c^{AF}; \mathbf{k}=0\}$.

**A333333:** $\xi_1(\mathbf{k})$ is between 0 ($\mathbf{k}=0$) and $\frac{9}{4}$ ($\mathbf{k}= \pm(\frac{2\pi}{3}, \frac{2\pi}{3})$). Hence $X_{\mathbf{k}}(v)$ is between $a_0^2(v) \geq 0$ and $a_0^2(v) + \frac{9}{4}a_1(v) \geq 0$. Last bound vanishes only at $v = -1$ while $a_0^2(v)$ vanishes only at $v=v_c$ (given in Tab. 2). Hence $X_{\mathbf{k}}(v) \geq 0$ and vanishes only at $\{v=-1; \mathbf{k}= \pm(\frac{2\pi}{3}, \frac{2\pi}{3})\}$ and $\{v=v_c; \mathbf{k}=0\}$.

# E   Calculations for the specific heat

In this section, we evaluate the integral $I_1(v)$ around $v_c$ and $I_{0,v_c}$ and $I_{2,v_c}$. These integrals are defined in Eqs. (24a).

## E.1   Expression of energy and specific heat

Eq. (19) states that $-\beta mf$ depends on $\beta$ only through $v = \tanh \beta J$. To get Eqs. (22) and (23), we start with intermediate functions, the derivatives of $-\beta mf$ with respect to $v$:

$$e_v = -\frac{d(-\beta mf)}{dv} = -\frac{n_l v}{1-v^2} - \frac{n_v}{1+v} - \frac{1}{2}I_0(v), \tag{E.1}$$

$$e'_v = -\frac{(1+v^2)n_l}{(1-v^2)^2} + \frac{n_v}{(1+v)^2} - \frac{I_1(v) - I_2(v)}{2}, \tag{E.2}$$

where $I_0$, $I_1$ and $I_2$ are given in Eq. (24a) and recalled below, and the prime indicates a derivative with respect to $v$.

$$I_0(v) = \int_{BZ} \frac{d^2\mathbf{k}}{4\pi^2} \frac{X'_{\mathbf{k}}(v)}{X_{\mathbf{k}}(v)}, \qquad I_1(v) = \int_{BZ} \frac{d^2\mathbf{k}}{4\pi^2} \frac{X''_{\mathbf{k}}(v)}{X_{\mathbf{k}}(v)}, \qquad I_2(v) = \int_{BZ} \frac{d^2\mathbf{k}}{4\pi^2} \left[ \frac{X'_{\mathbf{k}}(v)}{X_{\mathbf{k}}(v)} \right]^2. \tag{E.3}$$

Using $\frac{dv}{d\beta} = J(1-v^2)$, the energy $e$ and specific heat $c_V$ per site are given by:

$$\frac{me}{J} = -\frac{1}{J} \frac{d(-\beta mf)}{d\beta} = (1-v^2)e_v \tag{E.4}$$

$$= -n_l v - n_v(1-v) - \frac{1-v^2}{2}I_0(v),$$

$$\frac{mc_V}{\beta^2 J^2} = \frac{1}{J^2} \frac{d^2(-\beta mf)}{d\beta^2} = -(1-v^2)\left((1-v^2)e'_v - 2v e_v\right)$$

$$= (1-v^2)(n_l - n_v - v I_0(v)) + \frac{(1-v^2)^2}{2}(I_1(v) - I_2(v)). \tag{E.5}$$

## E.2   Singular and constant term of $I_1(v)$

We expand $X_{\mathbf{k}}(v)$ around the singularity as a function of $\epsilon_v = v - v_c$:

$$\frac{X_{\mathbf{k}}(v)}{a'_0(v_c)^2} = \epsilon_v^2 + \mathcal{S}_0(\mathbf{k}) + \mathcal{S}_1(\mathbf{k})\epsilon_v + \mathcal{S}_2(\mathbf{k})\epsilon_v^2 + o(\epsilon_v^2), \tag{E.6}$$

$$\mathcal{S}_\alpha(\mathbf{k}) = \sum_{i=1}^{n_f} \frac{a_i^{(\alpha)}(v_c)}{a'_0(v_c)^2 \, \alpha!} \xi_i(\mathbf{k}), \tag{E.7}$$

where $a_i^{(\alpha)}$ is the $\alpha$th derivative of $a_i$ with respect to $v$.

We define a function $I_1^{(s)}(v)$ with the same dominant term as $I_1(v)$, chosen as the right-hand integral of Eq. (29) and determine its behavior near $v_c$:

$$I_1^{(s)}(v_c + \epsilon_v) = \int_{BZ} \frac{d^2\mathbf{k}}{4\pi^2} \frac{2}{\epsilon_v^2 + \mathcal{S}_0^{(s)}(\mathbf{k})} = \int_{-\pi}^{\pi} \frac{dk_x}{\pi^2} \frac{\arctan \frac{\pi \mu_{yy} + k_x \mu_{xy}}{\tau}}{\tau},$$

where $\tau = \sqrt{\epsilon_v^2 \mu_{yy} + \delta k_x^2}$, $\mathcal{S}_0^{(s)}(\mathbf{k}) = \mu_{xx} k_x^2 + 2\mu_{xy} k_x k_y + \mu_{yy} k_y^2$. $\delta$ and $\mu_{ab}$ are defined in (30b) and Eq. (30c). If $\mu_{yy} \geq \mu_{xy}$, the argument of the arctan is positive for all $k_x \in [-\pi, \pi]$

and we use the identity $\arctan(x) = \frac{\pi}{2} - \arctan\frac{1}{x}$:

$$I_1^{(s)}(v_c + \epsilon_v) = \int_{-\pi}^{\pi} dk_x \left[ \frac{1}{2\pi\tau} - \frac{\arctan\frac{\tau}{\pi\mu_{yy}+k_x\mu_{xy}}}{\pi^2\tau} \right]$$

$$= \frac{\text{asinh}\frac{\pi\sqrt{\delta}}{|\epsilon_v|\sqrt{\mu_{yy}}}}{\pi\sqrt{\delta}} - \int_{-\pi}^{\pi} dk_x \frac{\arctan\frac{\sqrt{\delta}|k_x|}{\pi\mu_{yy}+k_x\mu_{xy}}}{\pi^2\sqrt{\delta}|k_x|} + o(1), \tag{E.8}$$

where the first term is exactly integrated while in the second we use $\lim_{\epsilon_v \to 0} \tau = \sqrt{\delta}|k_x|$. Using the dilogarithm function $\text{Li}_2(z) = -\int_0^z du \frac{\ln(1-u)}{u}$ and:

$$\mathcal{A}(a,b,p) = \int dp \frac{\arctan\frac{|p|}{a+bp}}{|p|}$$

$$= \Im\left( \text{Li}_2\left(\frac{ip}{\phi}\right) + \text{Li}_2\left(\frac{a(1+ib)}{(b^2+1)\phi}\right) \right) + \arctan\left(\frac{1}{b}\right)\ln\left(\frac{\phi}{a}\right), \tag{E.9}$$

where $\phi = bp + a$, we find:

$$I_1^{(s)}(v_c + \epsilon_v) = \frac{1}{\pi\sqrt{\delta}}\left( -\ln|\epsilon_v| + \ln\frac{2\pi\sqrt{\delta}}{\sqrt{\mu_{yy}}} \right) - \frac{\mathcal{A}(a,b,\pi) - \mathcal{A}(a,b,-\pi)}{\pi^2\sqrt{\delta}} + o(1), \tag{E.10}$$

with $a = \pi\mu_{yy}/\sqrt{\delta}$ and $b = \mu_{xy}/\sqrt{\delta}$. As expected, this formula is symmetrical by exchanging the indices $x$ and $y$. For the square lattice, with $\mu_{xy} = 0$ and $\sqrt{\delta} = \mu_{xx} = \mu_{yy}$, we find $\mathcal{A}(\pi, 0, \pm\pi) = \pm\text{Catalan}$, where the Catalan's constant is $\text{Catalan} = \sum_{n=0}^{\infty} \frac{(-1)^n}{(2n+1)^2} \simeq 0.915965594$.

We now evaluate $I_1(v_c + \epsilon_v) - I_1^{(s)}(v_c + \epsilon_v)$ in the limit $\epsilon_v \to 0$:

$$I_1(v_c + \epsilon_v) - I_1^{(s)}(v_c + \epsilon_v) = \int_{BZ} \frac{d^2\mathbf{k}}{4\pi^2} \left[ \frac{2(1+\mathcal{S}_2(\mathbf{k}))}{\epsilon_v^2 + \mathcal{S}_0(\mathbf{k})} - \frac{2}{\epsilon_v^2 + \mathcal{S}_0^{(s)}(\mathbf{k})} \right] + o(1)$$

$$\sim \int_{BZ} \frac{d^2\mathbf{k}}{4\pi^2} \left[ \frac{2\mathcal{S}_2(\mathbf{k})}{\mathcal{S}_0(\mathbf{k})} - \frac{2\bar{\mathcal{S}}_0(\mathbf{k})}{\mathcal{S}_0(\mathbf{k})\mathcal{S}_0^{(s)}(\mathbf{k})} \right], \tag{E.11}$$

where $\bar{\mathcal{S}}_0(\mathbf{k}) = \mathcal{S}_0(\mathbf{k}) - \mathcal{S}_0^{(s)}(\mathbf{k})$. Finally, around $v_c$ we obtain:

$$\mathcal{I}_1 = \lim_{\epsilon_v \to 0} I_1(v_c + \epsilon_v) + \pi/\sqrt{\delta}\ln|\epsilon_v|$$

$$= \frac{1}{\pi\sqrt{\delta}}\ln\frac{2\pi\sqrt{\delta}}{\sqrt{\mu_{yy}}} + \frac{\mathcal{A}(a,b,\pi) - \mathcal{A}(a,b,-\pi)}{\pi^2\sqrt{\delta}} + \int_{BZ} \frac{d^2\mathbf{k}}{4\pi^2} \frac{2\mathcal{S}_2(\mathbf{k})\mathcal{S}_0^{(s)} - 2\bar{\mathcal{S}}_0(\mathbf{k})}{\mathcal{S}_0(\mathbf{k})\mathcal{S}_0^{(s)}(\mathbf{k})}. \tag{E.12}$$

### E.3 $I_0(v_c)$ and $I_2(v_c)$

Keeping the leading terms of $I_0(v)$ and $I_2(v)$ we obtain successively:

$$I_0(v_c) = \int_{BZ} \frac{d^2\mathbf{k}}{4\pi^2} \frac{\mathcal{S}_1(\mathbf{k})}{\mathcal{S}_0(\mathbf{k})}, \tag{E.13}$$

$$I_2(v_c + \epsilon_v) \sim \int_{BZ} \frac{d^2\mathbf{k}}{4\pi^2} \left[ \frac{\mathcal{S}_1(\mathbf{k}) + 2\epsilon_v}{\mathcal{S}_0(\mathbf{k}) + \epsilon_v^2} \right]^2$$

$$\sim \int_{BZ} \frac{d^2\mathbf{k}}{4\pi^2} \left[ \frac{\mathcal{S}_1(\mathbf{k})}{\mathcal{S}_0(\mathbf{k})} \right]^2 + \int_{BZ} \frac{d^2\mathbf{k}}{4\pi^2} \left[ \frac{2\epsilon_v}{\epsilon_v^2 + \mathcal{S}_0(\mathbf{k})} \right]^2. \tag{E.14}$$

The second integral has significant contributions near $\mathbf{k} = 0$, where we replace $\mathcal{S}_0$ by $\mathcal{S}_0^{(s)}$:

$$\int_{BZ} \frac{d^2\mathbf{k}}{4\pi^2} \left( \frac{2\epsilon_\nu}{\epsilon_\nu^2 + \mathcal{S}_0^{(s)}(\mathbf{k})} \right)^2 = \frac{1}{\pi\sqrt{\delta}} + o(1). \tag{E.15}$$

Thus, in the limit $\epsilon_\nu \to 0$, we obtain:

$$I_2(\nu_c) = \int_{BZ} \frac{d^2\mathbf{k}}{4\pi^2} \left[ \frac{\mathcal{S}_1(\mathbf{k})}{\mathcal{S}_0(\mathbf{k})} \right]^2 + \frac{1}{\pi\sqrt{\delta}}. \tag{E.16}$$

### E.4 Exact results when $n_f = 1$

This concerns the lattices A4444, A333333, A666, A3636 and A3CC. $\xi_1(\mathbf{k})$ has the following simple expression, with the square lattice as a special case:

$$\xi_1(\mathbf{k}) = \sin^2 \frac{k_x}{2} + \sin^2 \frac{k_y}{2} \qquad (A4444), \tag{E.17a}$$

$$= \sin^2 \frac{k_x}{2} + \sin^2 \frac{k_y}{2} + \sin^2 \frac{k_x + k_y}{2} \qquad (A333333, A666, A3636, A3CC). \tag{E.17b}$$

An analytical expression is obtained for $c_V$ using a slightly different method than in previous subsections. With $u(\nu) = \frac{a_0^2(\nu)}{a_1(\nu)}$ and when $a_1(\nu) > 0$ (which is not true for AF-A333333), Eqs. (19) reads:

$$-m\beta f = m \ln 2 - \frac{n_l}{2} \ln(1-\nu^2) + \frac{\ln\left((1+\nu)^{2n_\nu} a_1(\nu)\right)}{2} + \frac{1}{2} \int_{BZ} \frac{d^2\mathbf{k}}{4\pi^2} \ln\left(u + \xi_1(\mathbf{k})\right). \tag{E.18}$$

The energy per site $e$ is:

$$\frac{me}{J} = -\nu n_l - (1-\nu)n_\nu - \frac{1-\nu^2}{2} \left( \frac{a_1'}{a_1} + u' H_1(u) \right), \tag{E.19}$$

with:

$$H_j(u) = \int_{BZ} \frac{d^2\mathbf{k}}{4\pi^2} \frac{1}{(u + \xi_1(\mathbf{k}))^j}. \tag{E.20}$$

At $\nu = \nu_c$, $u' H_1$ vanishes, thus the critical energy is:

$$\frac{me_c}{J} = -\nu_c n_l - (1-\nu_c)n_\nu - \frac{1-\nu_c^2}{2} \frac{a_1'(\nu_c)}{a_1(\nu_c)}. \tag{E.21}$$

Then $c_V$ becomes:

$$\frac{mc_V}{\beta^2 J^2} = (n_l - n_\nu)(1-\nu^2) + \frac{\frac{d^2 \ln a_1}{d(\beta J)^2} + \frac{d^2 u}{d(\beta J)^2} H_1(u) - \left(\frac{du}{d(\beta J)}\right)^2 H_2(u)}{2}. \tag{E.22}$$

For the square lattice we obtain:

$$H_1(u) = \frac{2}{\pi(u+1)} \mathcal{K}\left(\frac{1}{1+u}\right), \tag{E.23}$$

$$H_2(u) = \frac{2}{\pi u(u+2)} \mathcal{E}\left(\frac{1}{1+u}\right), \tag{E.24}$$

where $\mathcal{K}(z) = \int_0^1 dt/(\sqrt{1-t^2}\sqrt{1-z^2t^2}) = \frac{1}{2\pi}\int_0^\pi\int_0^\pi dxdy/(1+z\cos x\cos y)$ and $\mathcal{E}(z) = \int_0^1 dt\sqrt{1-z^2t^2}/\sqrt{1-t^2}$ are elliptic functions assuming $z \in [0,1]$.

For the other lattices with $n_f = 1$, we have:

$$H_1(u) = h_u\,\mathcal{K}(z_0)\,, \tag{E.25}$$

$$H_2(u) = \frac{h_u}{4u+9}\left[\mathcal{K}(z_0) + \frac{12(u+2)^2\mathcal{E}(z_0)}{(u_++1)(u_--1)}\right]\,, \tag{E.26}$$

with

$$h_u = \frac{4}{\pi\sqrt{u_+-1}\sqrt{u_-+1}}\,, \tag{E.27}$$

$$z_0 = \frac{\sqrt{2}\sqrt{u_+-u_-}}{\sqrt{u_+-1}\sqrt{u_-+1}}\,, \tag{E.28}$$

$$u_\pm = 2u+4\pm\sqrt{4u+9}\,. \tag{E.29}$$

Around the singularity, we have:

$$\frac{d^2u}{d(\beta J)^2} = 2u_{2c} + o(1)\,, \tag{E.30}$$

$$\left(\frac{du}{d(\beta J)}\right)^2 = 4u_{2c}u + o(u)\,, \tag{E.31}$$

$$u_{2c} = \frac{(1-v_c^2)^2 a_0'(v_c)^2}{a_1(v_c)}\,. \tag{E.32}$$

Thus we are left with the limit of $H_1(u) - 2u H_2(u)$ around $u = 0$. Noting that $\mathcal{E}(1) = 1$ and $\mathcal{K}(1-\epsilon) = -\frac{1}{2}\ln\frac{\epsilon}{8} + o(1)$, we obtain:

$$H_1(u) - 2u H_2(u) = c_1\left(-\ln\frac{u}{c_2} - 2\right) + o(1)\,, \tag{E.33}$$

with $c_1 = 1/\pi$, $c_2 = 8$ for the square lattice and $c_1 = 1/(\pi\sqrt{3})$, $c_2 = 18$ otherwise. Around the singularity, we have $u = u_{2c}\epsilon_x^2$ and:

$$\frac{mA}{\beta_c^2 J^2} = 2c_1 u_{2c}\,, \tag{E.34}$$

$$\frac{mB_x}{\beta_c^2 J^2} = (n_l - n_v)(1-v_c^2) - c_1 u_{2c}\left(\ln\frac{u_{2c}}{c_2} + 2\right) + \frac{1}{2}(1-v_c^2)\left[(1-v^2)(\ln(a_1(v)))'\right]'\Big|_{v_c}\,. \tag{E.35}$$

Tab. 4 shows the exact results for $n_f = 1$.

# F  Duality

The dual $\tilde{\mathcal{L}}$ of a graph $\mathcal{L}$ drawn on a surface (not necessarily plane) with non-intersecting links, is a graph where links are quarter turned or so, and faces become vertices and converse.

### F.1 Relationship between the Kac-Ward polynomials of dual lattices

The Kac-Ward polynomial $\bar{P}_\Lambda$ of a connected finite planar graph or $\bar{P}_{\hat{W}(0)}$ of a period of a connected double-periodic planar graph yields the Kac-Ward polynomial of its dual through:

$$\bar{P}_{\tilde{\Lambda}}(\tilde{v}) = 2^{1-\tilde{N}_s}(1+\tilde{v})^{N_l}\bar{P}_\Lambda\left(\frac{1-\tilde{v}}{1+\tilde{v}}\right), \tag{F.1}$$

$$\bar{P}_{\tilde{\hat{W}}(0)}(\tilde{v}) = 2^{-\tilde{m}}(1+\tilde{v})^{n_l}\bar{P}_{\hat{W}(0)}\left(\frac{1-\tilde{v}}{1+\tilde{v}}\right). \tag{F.2}$$

$\tilde{N}_s$ and $N_l$ are the numbers of faces and links of the finite planar graph. $\tilde{N}_s$ counts an outer face, which contains point at infinity added to plan to turn it into a sphere. $\tilde{m}$ and $n_l$ are the numbers of faces and links per period.

*Proof.* If $l' = c \to b$ and $l = a \to b$ are two clockwise consecutive oriented links leading to site $b$, then $l$ and $-l' = b \to c$ are two clockwise consecutive oriented links of a face. Let $\frac{l}{l'} = -i\Lambda_{l,-l'}$. It is the common value of $\frac{\Lambda_{l,b\to d}}{\Lambda_{l',b\to d}}$ for all $d$ but $a$ and $c$. We denote $\mathbf{e}_l$ the vectors that form a basis of the vector space on which matrix $\Lambda$ operates (its rows and columns are labeled by $l$). Hence

$$\Lambda\left(\mathbf{e}_l - \frac{l}{l'}\mathbf{e}_{l'}\right) = i\frac{l}{l'}\mathbf{e}_{-l'} + i\mathbf{e}_{-l}, \tag{F.3}$$
$$(I - v\Lambda)\left(\mathbf{e}_l - \frac{l}{l'}\mathbf{e}_{l'}\right) = \mathbf{e}_l - iv\mathbf{e}_{-l} - \frac{l}{l'}(\mathbf{e}_{l'} + iv\mathbf{e}_{-l'}).$$

Let $R : \mathbf{e}_l \mapsto \mathbf{e}_l + i\mathbf{e}_{-l}$. Hence $\det R = 2^{N_l}$ since $\det\begin{pmatrix} 1 & i \\ i & 1 \end{pmatrix} = 2$. Let $\tilde{v} = \frac{1-v}{1+v}$. Hence $v = \frac{1-\tilde{v}}{1+\tilde{v}}$.

$$(1+\tilde{v})(I-v\Lambda)\left(\mathbf{e}_l - \frac{l}{l'}\mathbf{e}_{l'}\right) = (1+\tilde{v})\mathbf{e}_l - i(1-\tilde{v})\mathbf{e}_{-l} - \frac{l}{l'}((1+\tilde{v})\mathbf{e}_{l'} + i(1-\tilde{v})\mathbf{e}_{-l'}) \tag{F.4}$$

$$= R\left(\tilde{v}\mathbf{e}_l - i\mathbf{e}_{-l} - \frac{l}{l'}(\mathbf{e}_{l'} - i\tilde{v}\mathbf{e}_{-l'})\right). \tag{F.5}$$

For Archimedan lattices, faces are regular polygons, and links of the dual lattice are obtained by turning all links by $+\frac{\pi}{2}$. Then $\tilde{l}'$ and $-\tilde{l}$ are clockwise consecutive oriented links on $\tilde{\mathcal{L}}$. Previous calculations work when replacing $l, l', v, \Lambda$ and $\frac{l}{l'}$ with $\tilde{l}', -\tilde{l}, \tilde{v}, \tilde{\Lambda}$ and $\frac{\tilde{l}'}{-\tilde{l}} = -i\frac{l'}{l} = -i/\frac{l}{l'}$. Eq. (F.3) becomes:

$$(I - \tilde{v}\tilde{\Lambda})\left(\mathbf{e}_{\tilde{l}'} + i\frac{l'}{l}\mathbf{e}_{-\tilde{l}}\right) = \mathbf{e}_{\tilde{l}'} - i\tilde{v}\mathbf{e}_{-\tilde{l}'} + i\frac{l'}{l}\left(\mathbf{e}_{-\tilde{l}} + i\tilde{v}\mathbf{e}_{\tilde{l}}\right), \tag{F.6}$$

$$(I - \tilde{v}\tilde{\Lambda})\left(-\frac{l}{l'}\mathbf{e}_{\tilde{l}'} - i\mathbf{e}_{-\tilde{l}}\right) = \tilde{v}\mathbf{e}_{\tilde{l}} - i\mathbf{e}_{-\tilde{l}} - \frac{l}{l'}\left(\mathbf{e}_{\tilde{l}'} - i\tilde{v}\mathbf{e}_{-\tilde{l}'}\right). \tag{F.7}$$

Equations (F.5) and (F.7) have very similar right-hand side and prove that

$$TR^{-1}(1+\tilde{v})(I-v\Lambda)M = (I - \tilde{v}\tilde{\Lambda})\tilde{M}ST,$$

where

$$M : \mathbf{e}_l \mapsto \mathbf{e}_l - \frac{l}{l'}\mathbf{e}_{l'}, \qquad \tilde{M} : \mathbf{e}_{\tilde{l}'} \mapsto \mathbf{e}_{\tilde{l}'} - \frac{\tilde{l}'}{-\tilde{l}}\mathbf{e}_{-\tilde{l}}, \qquad T : \mathbf{e}_l \mapsto \mathbf{e}_{\tilde{l}}, \qquad S : \mathbf{e}_{\tilde{l}} \mapsto -\frac{l}{l'}\mathbf{e}_{\tilde{l}'},$$

$$\det M = \prod_{b\text{ site}}\left(1 - \prod_{l=a\to b}\frac{l}{l'}\right) = \prod_{b\text{ site}}2 = 2^{N_s}, \qquad \det\tilde{M} = 2^{\tilde{N}_s}, \qquad \det S = 1.$$

Hence

$$
\begin{aligned}
\det((1+\tilde{v})(I-v\Lambda)M) &= \det(R(I-\tilde{v}\tilde{\Lambda})\tilde{M}S), \\
(1+\tilde{v})^{2N_l}P(v)2^{N_s} &= 2^{N_l}\tilde{P}(\tilde{v})2^{\tilde{N}_s}, \\
(1+\tilde{v})^{N_l}\bar{P}\left(\frac{1-\tilde{v}}{1+\tilde{v}}\right) &= 2^{(N_l+\tilde{N}_s-N_s)/2}\bar{\tilde{P}}(\tilde{v}).
\end{aligned}
\tag{F.8}
$$

The Euler-Poincaré characteristic of a sphere is 2, meaning that $N_s - N_l + \tilde{N}_s = 2$ for a planar connected graph. Then $2^{(N_l+\tilde{N}_s-N_s)/2} = 2^{\tilde{N}_s-1}$. But the Euler-Poincaré characteristic of a torus is 0, meaning that $m - n_l + \tilde{m} = 0 \, (= 2/\infty)$ for a mesh of a decorated lattice i.e. a period of a planar connected double-periodic graph. Then $2^{(N_l+\tilde{N}_s-N_s)/2}$ is to be replaced with $2^{(n_l+\tilde{m}-m)/2} = 2^{\tilde{m}}$. □

This proof still works if the initial graph is not Archimedean. Then dual links are not exactly orthogonal to initial links. But $\det(I-v\Lambda)$ does not change when directions of links are slightly changed.

Eq. (43) is the direct consequence of Eq. (F.8), but the proof of Eq. (F.8) works also with $\mathbf{k} \neq 0$ and proves also Eq. (44).

### F.2 Kac-Ward polynomial and duality for finite planar ferromagnetic graph

In this section, we give another proof, the one used historically, relating a F model on a finite graph at high temperature with the F model on the dual graph at low temperature.

To get an Ising model on the graph $\tilde{\mathcal{L}}$, we can keep the graph $\mathcal{L}$ and put a spin on each face, instead of each vertex of $\mathcal{L}$. Then an even subgraph of $\mathcal{L}$ can be viewed as the borderline between the set of faces of spin 1 and the set of faces of spin -1. If $\mathcal{L}$ is planar and $v = e^{-2\tilde{\beta}\tilde{J}}$ then $2v^{-N_l/2}\sum v^r g_r$ is the partition fonction of $\tilde{\mathcal{L}}$. The factor 2 is here because the borderline is the same when flipping the spins of all faces. We have $v = \tanh\beta J = e^{-2\tilde{\beta}\tilde{J}}$. For $\tilde{\mathcal{L}}$, we have $\tilde{v} = \tanh\tilde{\beta}\tilde{J} = e^{-2\beta J}$ since $\tilde{v} = \frac{1-v}{1+v}$ and $v = \frac{1-\tilde{v}}{1+\tilde{v}}$. Then $v, \tilde{v} \in [0,1]$. This gives the last part of:

$$
Z2^{-N_s}(1-v^2)^{N_l/2} = \sum_{r=0}^{N_l} v^r g_r = \sqrt{P(v)} = \frac{1}{2}v^{N_l/2}\tilde{Z}.
\tag{F.9}
$$

Eq. (7) and (8) give the first and middle part of it. The dual of this equation is:

$$
\tilde{Z}2^{-\tilde{N}_s}(1-\tilde{v}^2)^{N_l/2} = \sum_{r=0}^{N_l} \tilde{v}^r \tilde{g}_r = \sqrt{\tilde{P}(\tilde{v})} = \frac{1}{2}\tilde{v}^{N_l/2}Z.
\tag{F.10}
$$

Equating $\tilde{Z}$ from these two equations gives:

$$
\frac{\sqrt{\tilde{P}(\tilde{v})}}{\sqrt{P(v)}} = 2^{1-\tilde{N}_s}\left(\frac{1-\tilde{v}^2}{v}\right)^{N_l/2} = 2^{1-\tilde{N}_s}(1+\tilde{v})^{N_l}.
\tag{F.11}
$$

Equating $Z$ gives an inverse ratio equal to $2^{1-N_s}(1+v)^{N_l}$. But $(1+v)(1+\tilde{v}) = 2$ and the product of both ratios is $2^{2-N_s+N_l-\tilde{N}_s} = 2^0$ as due. Similarly, Eq. (F.9) gives

$$
\frac{Z}{\tilde{Z}} = 2^{N_s-1-N_l/2}\left(\frac{2v}{1-v^2}\right)^{N_l/2} = 2^{(N_s-\tilde{N}_s)/2}\sinh^{N_l/2}2\beta J,
\tag{F.12}
$$

which emphasizes that $\sinh 2\beta J$ is the inverse of $\sinh 2\tilde{\beta}\tilde{J}$. However $\frac{2v}{1-v^2}\frac{2\tilde{v}}{1-\tilde{v}^2} = 1$ is equivalent to $\tilde{v} = \frac{1-v}{1+v}$ or $\tilde{v} = \frac{v+1}{v-1}$. This is why, in this paper, we always use $\tilde{v} = \frac{1-v}{1+v}$ rather than $\sinh 2\beta J \sinh 2\tilde{\beta}\tilde{J} = 1$, to avoid the wrong value $\tilde{v} = \frac{v+1}{v-1}$.

All of this is a simpler way to get Eq. (F.1). But this way we cannot directly get Eq. (F.2), because the middle and last part of Eq. (F.9) need a planar graph: A loop on a flat torus which crosses once the right side does not part the torus in two zones.

So far, we assumed all $v_l$ are equal to $v$. This is not mandatory. For instance in Eq. (F.11) we may replace $(1 + \tilde{v})^{N_l}$ by $\prod_l (1 + \tilde{v}_l)$.

# G  Derivation of duality relation for some thermodynamic quantities in F models

In this section, we detail the results given in Sec. 6.1.

The function $\tilde{a}_0$ is deduced from Eq. (43), $\tilde{a}_i(\tilde{v})/\tilde{a}_0^2(\tilde{v}) = a_i(v)/a_0^2(v)$ and $\tilde{\xi}_i(\mathbf{k}) = \xi_i(\mathbf{k})$.

From the relation of the free energies of the lattice $\mathcal{L}$ and its dual $\tilde{\mathcal{L}}$, given in Eq.(46), we deduce the relations between the energy and specific heat in $\mathcal{L}$ and $\tilde{\mathcal{L}}$. The energy per site $e(v)$ is:

$$
\begin{aligned}
\frac{me(v)}{J} &= -\frac{1}{J}\frac{\partial(-\beta mf)}{\partial\beta} \\
&= -\frac{1}{J}\frac{\partial(-\beta mf)}{\partial v}\frac{\partial v}{\partial\beta} = -\frac{\partial(-\beta mf)}{\partial v}(1-v^2).
\end{aligned}
\tag{G.1}
$$

Similarly, we have

$$
\begin{aligned}
\frac{\tilde{m}\tilde{e}(\tilde{v})}{J} &= -\frac{1}{J}\frac{\partial(-\beta\tilde{m}\tilde{f})}{\partial\tilde{\beta}} \\
&= -\frac{\partial(-\beta\tilde{m}\tilde{f})}{\partial\tilde{v}}(1-\tilde{v}^2) = 2v\frac{\partial(-\beta\tilde{m}\tilde{f})}{\partial v} \\
&= \frac{-2v}{1-v^2}\frac{me(v)}{J} - n_l\frac{1+v^2}{1-v^2},
\end{aligned}
\tag{G.2}
$$

that we rewrite in a symmetric way as

$$
\frac{\tilde{m}}{1-\tilde{v}}\left(\frac{\tilde{e}}{J}+\frac{1+\tilde{v}^2}{2\tilde{v}}\right) + \frac{m}{1-v}\left(\frac{e}{J}+\frac{1+v^2}{2v}\right) = 0.
\tag{G.3}
$$

The specific heat per site $c_V$ is given by

$$
\begin{aligned}
\frac{mc_V}{\beta^2 J^2} &= \frac{1}{J^2}\frac{\partial^2(-\beta mf)}{\partial\beta^2} \\
&= \frac{\partial^2(-\beta mf)}{\partial v^2}(1-v^2)^2 - 2v(1-v^2)\frac{\partial(-\beta mf)}{\partial v},
\end{aligned}
\tag{G.4}
$$

and

$$
\begin{aligned}
\frac{\tilde{m}\tilde{c}_V}{\tilde{\beta}^2 J^2} &= \frac{\partial^2(-\beta\tilde{m}\tilde{f})}{\partial\tilde{v}^2}(1-\tilde{v}^2)^2 - 2\tilde{v}(1-\tilde{v}^2)\frac{\partial(-\beta\tilde{m}\tilde{f})}{\partial\tilde{v}} \\
&= 4v^2\frac{\partial^2(-\beta\tilde{m}\tilde{f})}{\partial v^2} + 4v\frac{\partial(-\beta\tilde{m}\tilde{f})}{\partial v} \\
&= 4v^2\frac{\partial^2(-\beta mf)}{\partial v^2} + 4v\frac{\partial(-\beta mf)}{\partial v} - \frac{8v^2 n_l}{(1-v^2)^2} \\
&= \frac{4v^2}{(1-v^2)^2}\left(\frac{mc_V}{\beta^2 J^2} - \frac{1+v^2}{v}\frac{me}{J} - 2n_l\right),
\end{aligned}
\tag{G.5}
$$

that we rewrite in a symmetric way as

$$
\frac{2\tilde{v}}{1-\tilde{v}^2}\frac{\tilde{m}\tilde{c}_V}{\tilde{\beta}^2 J^2} - \frac{1+\tilde{v}^2}{1-\tilde{v}^2}\frac{\tilde{m}\tilde{e}}{J} + \frac{\tilde{m}}{2\tilde{v}}\frac{\tilde{v}^4 - 6\tilde{v}^2 + 1}{1-\tilde{v}^2} = \frac{2v}{1-v^2}\frac{mc_V}{\beta^2 J^2} - \frac{1+v^2}{1-v^2}\frac{me}{J} + \frac{m}{2v}\frac{v^4 - 6v^2 + 1}{1-v^2}.
\tag{G.6}
$$

As expected, we recover the two correct limits for $\beta \to 0$ and $\beta \to \infty$. When $\beta = 0$ (thus $v = 0$ and $\tilde{v} = 1$), $e(0) = 0$, leading to and $\tilde{e}(1) = -Jn_l/\tilde{m}$. Expanding the right-hand side of Eq.(49), we find $\frac{\tilde{c}_V(1)}{\tilde{\beta}^2} = 0$. Reciprocally, when $\beta \to \infty$ (thus $v \to 1$ and $\tilde{v} \to 0$), $e(1) = -Jn_l/m$. Then, factorizing $n_l$, we find $\tilde{e}(\tilde{v} \to 0) \sim -(1-v)n_l/2 \to 0$. $\frac{c_V(1)}{\beta^2} = 0$ leading to $\frac{\tilde{c}_V(0)}{\tilde{\beta}^2 J^2} = n_l$. For the self-dual square lattice, we easily deduce $e_c = -J\sqrt{2}$.

Around the transition, we have:

$$\frac{\tilde{m}\tilde{A}}{\tilde{\beta}_c^2 J^2} = \frac{4v_c^2}{\left(1-v_c^2\right)^2} \frac{mA}{\beta_c^2 J^2} \,, \tag{G.7}$$

that we rewrite in a symmetric way as

$$\frac{2\tilde{v}_c}{1-\tilde{v}_c^2} \frac{\tilde{m}\tilde{A}}{\tilde{\beta}_c^2 J^2} = \frac{2v_c}{1-v_c^2} \frac{mA}{\beta_c^2 J^2} \,. \tag{G.8}$$

For the relation of $B$ we have to account that $\tilde{\epsilon} \neq \epsilon$:

$$\tilde{\epsilon} = 1 - \frac{\beta}{\tilde{\beta}_c} \sim \frac{\tilde{v}-\tilde{v}_c}{(1-\tilde{v}_c^2)\tilde{\beta}_c J} \sim -\frac{v-v_c}{2v_c\tilde{\beta}_c J} \sim -\frac{(1-v_c^2)\beta_c J}{2v_c\tilde{\beta}_c J}\epsilon \,. \tag{G.9}$$

The term $-\tilde{A}\ln\tilde{\epsilon}$ gives the constant contribution:

$$-\frac{\tilde{m}\tilde{A}}{\tilde{\beta}_c^2 J^2} \ln|\tilde{\epsilon}| = -\frac{\tilde{m}\tilde{A}}{\tilde{\beta}_c^2 J^2} \ln|\epsilon| - \frac{\tilde{m}\tilde{A}}{\tilde{\beta}_c^2 J^2} \ln\frac{(1-v_c^2)\beta_c J}{2v_c\tilde{\beta}_c J} \,. \tag{G.10}$$

Finally

$$\frac{\tilde{m}\tilde{B}}{\tilde{\beta}_c^2 J^2} = \frac{4v_c^2}{\left(1-v_c^2\right)^2}\left(\frac{mB}{\beta_c^2 J^2} - \frac{1+v_c^2}{v_c}\frac{me_c}{J} - 2n_l + \frac{mA}{\beta_c^2 J^2}\ln\frac{(1-v_c^2)\beta_c}{2v_c\tilde{\beta}_c}\right), \tag{G.11}$$

that we rewrite in a symmetric way as

$$\begin{aligned}
\frac{2\tilde{v}}{1-\tilde{v}_c^2}\frac{\tilde{m}\tilde{B}}{\tilde{\beta}^2 J^2} &- \frac{1+\tilde{v}_c^2}{1-\tilde{v}_c^2}\frac{\tilde{m}\tilde{e}}{J} + \frac{\tilde{m}}{2\tilde{v}_c}\frac{\tilde{v}_c^4 - 6\tilde{v}_c^2 + 1}{1-\tilde{v}_c^2} + \frac{1}{2}A\ln\frac{(1-\tilde{v}_c^2)\tilde{\beta}_c^2}{2\tilde{v}_c} \\
&= \frac{2v_c}{1-v_c^2}\frac{mB}{\beta^2 J^2} - \frac{1+v_c^2}{1-v_c^2}\frac{me}{J} + \frac{m}{2v_c}\frac{v_c^4 - 6v_c^2 + 1}{1-v_c^2} + \frac{1}{2}A\ln\frac{(1-v_c^2)\tilde{\beta}_c^2}{2v_c} \,.
\end{aligned} \tag{G.12}$$

# H Derivation of the star-triangle transformation

The energies $e_T$ of the triangle with exchanges $J_1$, $J_2$ and $J_3$ and $e_S$ of the star of exchanges $K_1$, $K_2$ and $K_3$ (see Fig. 9a) are:

$$e_T = -J_1\sigma_2\sigma_3 - J_2\sigma_3\sigma_1 - J_3\sigma_1\sigma_2 \,, \tag{H.1}$$
$$e_S = -K_1\sigma_1\sigma_0 - K_2\sigma_2\sigma_0 - K_3\sigma_3\sigma_0 \,. \tag{H.2}$$

Replacing $e_T$ by $e_S$, the partition function is multiplied by $r$ if:

$$r\sqrt{\frac{\tilde{s}_1\tilde{s}_2\tilde{s}_3}{\tilde{t}_1\tilde{t}_2\tilde{t}_3}} = \frac{\tilde{s}_1\tilde{s}_2\tilde{s}_3 + 1}{1} = \frac{\tilde{s}_2\tilde{s}_3 + \tilde{s}_1}{\tilde{t}_2\tilde{t}_3} = \frac{\tilde{s}_3\tilde{s}_1 + \tilde{s}_2}{\tilde{t}_3\tilde{t}_1} = \frac{\tilde{s}_1\tilde{s}_2 + \tilde{s}_3}{\tilde{t}_1\tilde{t}_2} \,, \tag{H.3}$$

where $\tilde{t}_i = \exp(-2\beta J_i)$ and $\tilde{s}_i = \exp(-2\beta K_i)$. Solving these equations for $t_i$ gives:

$$\tilde{t}_1 = \sqrt{\frac{(\tilde{s}_3 \tilde{s}_1 + \tilde{s}_2)(\tilde{s}_1 \tilde{s}_2 + \tilde{s}_3)}{(\tilde{s}_2 \tilde{s}_3 + \tilde{s}_1)(\tilde{s}_1 \tilde{s}_2 \tilde{s}_3 + 1)}} \,. \tag{H.4}$$

A cyclic permutation of the indices in Eq. (H.4) gives formulae for $\tilde{t}_2$ and $\tilde{t}_3$. To invert these relations, we first set:

$$F(\epsilon_1, \epsilon_2, \epsilon_3) = 1 + \epsilon_1 \tilde{t}_2 \tilde{t}_3 + \epsilon_2 \tilde{t}_3 \tilde{t}_1 + \epsilon_3 \tilde{t}_1 \tilde{t}_2 = \frac{(\tilde{s}_1 + \epsilon_1)(\tilde{s}_2 + \epsilon_2)(\tilde{s}_3 + \epsilon_3)}{\tilde{s}_1 \tilde{s}_2 \tilde{s}_3 + 1} \,, \tag{H.5}$$

for $\epsilon_1 = \pm 1$, $\epsilon_2 = \pm 1$ and $\epsilon_3 = \epsilon_1 \epsilon_2$. Hence

$$\frac{1 - \tilde{s}_1}{1 + \tilde{s}_1} = \sqrt{\frac{F(-1,-1,+1)F(-1,+1,-1)}{F(+1,-1,-1)F(+1,+1,+1)}} \,, \tag{H.6}$$

and similarly for $\tilde{s}_2$ and $\tilde{s}_3$.

Eq. (H.6) is Eq. (H.4) where $\tilde{s}_i$ is replaced by $t_i$ and $\tilde{t}_i$ by $s_i$ with

$$t_i = \frac{1 - \tilde{t}_i}{1 + \tilde{t}_i} = \tanh \beta J_i \,, \tag{H.7}$$

$$s_i = \frac{1 - \tilde{s}_i}{1 + \tilde{s}_i} = \tanh \beta K_i \,, \tag{H.8}$$

since the star is the dual of the triangle.

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
