# Peer review of "Derivation of the free energy, entropy and specific heat for planar Ising models: Application to Archimedean lattices and their duals"

_SciPost Physics, doi:SciPost Phys. 19, 025 (2025)_

## Round 1 · Referee Report · Anonymous (Referee 1) · 2025-4-16

Report
This manuscript presents a systematic and detailed treatment of the exact thermodynamic quantities for the two-dimensional Ising model defined on Archimedean lattices and their dual (Laves) lattices. The authors employ an elegant and generalizable method based on the Kac–Ward determinantal formalism, which allows them to compute both analytical and numerical results for the free energy, energy, entropy, and specific heat over the full temperature range. A particularly original contribution is the systematic calculation of the coefficients A and B characterizing the logarithmic singularity of the specific heat at the critical temperature—quantities which were previously unknown for most of the considered lattices.
The manuscript offers a comprehensive and unified treatment of a wide range of exact results that were previously scattered across the literature. In my view, one of its major strengths lies in the consolidation, completion, and coherent presentation of these results. The tables summarizing critical temperatures, residual entropies, and coefficients of the specific heat singularity represent a practically new reference catalogue, which, to my knowledge, had not previously been available in such a complete and organized form.
The authors correctly distinguish between bipartite and frustrated lattices in the antiferromagnetic case, and the results they obtain are consistent with known values (e.g., ground-state energies and residual entropies), which supports the reliability and correctness of their computations.
I have only a few relatively minor comments regarding the manuscript: 1) Incomplete referencing: Although the authors cite many relevant works, they omit several key references that should definitely be acknowledged. These include: -Alessandro Codello (2010), J. Phys. A: Math. Theor. 43, 385002; -I. Syozi (1972), in Phase Transitions and Critical Phenomena, Vol. 1, eds. Domb & Green; -J. Strečka (2006), Physics Letters A 349, 505–508. In addition, it may be helpful to cite one or more overview articles that collect a broad set of exact results for Ising models on Archimedean and related lattices. For example: -C. Domb, Adv. Phys. 9 (1960) 149; -K.Y. Lin, Chinese J. Phys. 30 (1992) 287; -J. Strečka, M. Jaščur, Acta Phys. Slovaca 65 (2015) 235. These reviews contain extensive bibliographies (e.g., references [366–394] in Strečka & Jaščur) that cover many exact results relevant to the present work, including those not cited directly here. Referring to them would not only strengthen the contextual background, but also offer a more complete historical perspective. Moreover, Domb’s review contains explicit numerical values of critical amplitudes for the square, triangular and hexagonal lattices, which could serve as useful points of comparison in the discussion of the authors’ own amplitude results.
2) In the early sections, the authors introduce several interrelated mathematical objects and symbols but do so purely textually. I recommend including a simple schematic diagram with a clear caption that visually illustrates these concepts. Such an illustration would greatly help readers navigate the formalism, especially those less familiar with this type of combinatorial approach, and would enhance the pedagogical clarity of the paper.
3) Appendix references: Unless I overlooked something, the order in which appendices are cited in the main text appears inconsistent. I recommend checking and, if needed, adjusting the order or adding clarifying remarks. 4) Ambiguous formulation regarding the sign of J: Below Equation (6), the text states “For F (resp. AF) interactions (J>0)...” This is ambiguous and potentially confusing. I suggest rewriting it more explicitly as: “ferromagnetic (J>0), antiferromagnetic (J<0).”
5) Figure 2 caption: The caption mentions “blue links,” but no blue edges are visible in the figure. Please check whether this is a leftover from an earlier version, or clarify what is meant.
6) Figure 5 caption: In the last sentence, the caption refers to lattice “A3334,” which does not appear elsewhere in the paper. This seems to be a typographical error; presumably it should read “A33344.”
7) Punctuation after equations: The manuscript contains several cases of inconsistent punctuation following displayed equations—sometimes a period or comma is missing, other times a full stop is used even though the sentence continues. I recommend reviewing the manuscript for consistent punctuation to improve clarity and readability.
8) Sentences beginning with variables: In many places, sentences begin with expressions such as f(T), c_v , etc., which visually appear to begin with a lowercase letter. While this is not technically incorrect in scientific writing, I suggest rephrasing such sentences (e.g., “The function f(T)...”) to ensure smoother readability and presentation.
9) Applicability to other lattices – open question: It would be valuable if the authors briefly commented in the conclusion on the potential applicability of their approach beyond Archimedean and Laves lattices. For example, can similar methods be applied to other periodic planar graphs with non-uniform vertex degree? Is constant coordination number a necessary condition? Even a short discussion on this point would benefit readers considering extensions of this work.
Overall, this is a strong, original, and well-executed article. The results are consistent with existing literature and provide a wealth of systematically organized data that had previously been unavailable in unified form. The manuscript contains a few minor formal issues (punctuation, citation gaps, some unclear references) that should be corrected before publication.
In my opinion, this manuscript meets the criteria for publication in SciPost Physics. I recommend acceptance after minor revisions to address the points outlined above.
Recommendation
Ask for minor revision

---

## Round 1 · Referee Report · Anonymous (Referee 2) · 2025-4-22

Strengths
1- The manuscript provides a comprehensive overview of exact results for nearest-neighbor, translation-invariant Ising models on Archimedean lattices and their dual, gathering and completing results otherwise scattered across a number of papers and several decades of literature, thus providing a highly valuable reference. 2- It contributes the original and systematic calculation of the A and B terms controlling the logarithmic divergence of the specific heat for all ferromagnetic models on Archimedean and Laves lattices. 3- It benefits from a significant effort to make the main points of the calculations digestible while leaving the detailed proofs in appendices. 4- The authors highlight where specific features of the lattices are required (e.g., having Archimedean lattices, translation-invariant lattices) for the proofs to hold, facilitating generalization.
Weaknesses
1- In this version, a few references are missing.
Report
Particular focus and effort is put on (1) presenting the method in a digestible way, with detailed proofs relegated in appendix, (2) obtaining rigorous results in the case of periodic boundary conditions (where the graph is not technically speaking "planar") and (3) contributing original, exact results regarding the dominant and subdominant terms controlling the logarithmic divergence of the specific heat for ferromagnetic models.
The results are sound and, where comparable, consistent with the literature. The work is both pedagogical and contributes new original results, and the manuscript is well-written. Besides its original contributions, I believe it will be extremely helpful as a reference for future research: (1) It is a very helpful guide for the analytical calculations, pointing out where specific properties of the lattices are required for proofs to hold, and (2) it contributes important overview tables, which were not previously available in the literature in such complete form, with the main results for all Archimedean and Laves lattices.
Except for the requested minor changes below and in particular the missing references, my opinion is that the paper meets the acceptance criteria for publication in SciPost physics, and I recommend acceptance after minor revisions.
Requested changes
Warnings issued while processing user-supplied markup:
- Inconsistency: Markdown and reStructuredText syntaxes are mixed. Markdown will be used.
Add "#coerce:reST" or "#coerce:plain" as the first line of your text to force reStructuredText or no markup.
You may also contact the helpdesk if the formatting is incorrect and you are unable to edit your text.
1- Minor changes
a- References : while the main references are acknowledged and mentioned, a few references are still missing. As they are already mentioned in the other report, let me only add this one : - I.Syôzi, Statistics of Kagomé Lattice, Progress of Theoretical Physics, Volume 6, Issue 3, 1951, Pages 306–308 b- Fig. 2 is missing some of the colors mentioned in its caption. I believe these would be extremely helpful to clarify the main text as the results are not completely trivial.
2- Typos etc.:
I recommend that the authors carefully recheck the manuscript for typos (understandable given the length of the manuscript). In particular I have doubts here : a- It is unclear to me what u(\check{l}f) means in Eq. (10). Is this correct ? b- The reference to Eq. (19) at the beginning of paragraph (2.3) seems a bit strange (albeit perhaps correct) since it is only presented much later on in the paper. c- Above Eq. (18), I suspect P_0(1) should either have an overline or should be set equal to -2^{2 \tilde{m}}. d- In Eq. (18) I have a doubt: should it not be n ? e- There is a typo in the residual entropy for A3636 in sec. 5.2 (0.0518 should be 0.5018), as stated correctly in the related table.
3- "Proof"details I stumbled onto:
- I believe "insure" should be "ensure" in both cases.
- bottom of P. 6 "we fall is the situation"-> "we fall in the situation".
4- Optional changes / questions for the authors' consideration:
a- The angle \alpha is introduced quite early in the text but mostly used around Eq. (9). It could be helpful to the reader to only introduce it there and to explicitly mention it in Appendix A. b- The appendices may be easier to navigate if presented in the same order as the text, or alternatively if the choice of their structure is explained in the introduction or at the very beginning of the appendices. c- The various geometrical quantities introduced in the first sections can be hard to follow. A figure summarizing their different roles might be helpful, or an "intuition" behind their impact on the calculations e.g. above Eq. 9 and e.g. in the very beginning of sec. 2.4 when mentioning Eq. (70). c - The square torus is used to discuss periodic boundary conditions. In numerics, people sometimes use other constructions for tori to respect the lattice symmetries (e.g. in the triangular lattice case). May the author comment on whether this could affect their results ? d- The authors comment in the conclusion about the appearance of geometric characteristic playing an important role in the statistical mechanics of the Ising model. I may have missed it : I am curious whether their systematic analysis (see Table 1 & 3) provides further insight on possible systematic correlations between e.g. the value of n_nu and A, B. e- The authors may consider summarizing in the conclusions the directions they already pointed out where their results may or may not be generalizable, possible references where this might have been considered, and perhaps comment on a few other possible directions. For instance, is there a way to use the combinatorial method to charaterize the magnetization or the correlations (despite the absence of a solution in the presence of a magnetic field), and for instance identify in the AF model the presence of a zero-temperature critical point? Would the approach be applicable to spin glasses (e.g. J.F. Valdés, W. Lebrecht, E.E. Vogel, Physica A: Statistical Mechanics and its Applications, Volume 391, (2012))? Would it be applicable to amorphous lattices ? These last few questions are mostly out of curiosity inspired by this work.
Recommendation
Ask for minor revision

---

## Round 2 · Referee Report · Anonymous (Referee 1) · 2025-6-6

Report

I have reviewed the revised manuscript and the authors’ response. All my comments have been addressed clearly and satisfactorily. In addition, the authors implemented several thoughtful improvements beyond what was requested—for example, historical references and discovery dates in Table 1—which enhance the manuscript’s usefulness as a reference.

I am fully satisfied with the revision and recommend the manuscript for publication.

Recommendation

Publish (easily meets expectations and criteria for this Journal; among top 50%)

---

## Round 2 · Referee Report · Anonymous (Referee 2) · 2025-6-17

Report

The Authors have carefully considered my and the other referee's comments, implemented the necessary modification and improved their manuscript beyond our requests. In particular, I find the new Fig. 2 very helpful.

I can now highly recommend this manuscript for publication.

Recommendation

Publish (easily meets expectations and criteria for this Journal; among top 50%)

---

## Round 2 · Author Response

Warnings issued while processing user-supplied markup:

  • Inconsistency: Markdown and reStructuredText syntaxes are mixed. Markdown will be used.
    Add "#coerce:reST" or "#coerce:plain" as the first line of your text to force reStructuredText or no markup.
    You may also contact the helpdesk if the formatting is incorrect and you are unable to edit your text.

ANSWER TO REPORT 1

We thank the referee for his/her positive evaluation. We answer below to the requests for changes :

"1) Incomplete referencing "

We have added references to several reviews in the introduction, including the three mentionned by the referee: -C. Domb, Adv. Phys. 9 (1960) 149; -K.Y. Lin, Chinese J. Phys. 30 (1992) 287; -J. Strečka, M. Jaščur, Acta Phys. Slovaca 65 (2015) 235.

and in Table 1, we have added the year and reference where the critical temperature was first derived for each archimedean lattice, which led us to cite: -Alessandro Codello (2010), J. Phys. A: Math. Theor. 43, 385002; -I. Syozi (1972), in Phase Transitions and Critical Phenomena, Vol. 1, eds. Domb & Green; The reference 'J. Strečka (2006), Physics Letters A 349, 505–508' now appears in Sec.6, as it describe a star-triangle transformation between two lattices with different values of J on the links of two Archimedean lattices.

"2) In the early sections, the authors introduce several interrelated mathematical objects and symbols but do so purely textually. I recommend including a simple schematic diagram with a clear caption that visually illustrates these concepts. Such an illustration would greatly help readers navigate the formalism, especially those less familiar with this type of combinatorial approach, and would enhance the pedagogical clarity of the paper."

We have added a figure illustrating several notations (new Fig.2) used in sec. 2.1 and 2.2.

"3) Appendix references: Unless I overlooked something, the order in which appendices are cited in the main text appears inconsistent. I recommend checking and, if needed, adjusting the order or adding clarifying remarks."

We have reorganized the appendices so that they now appear in the same order as their references in the text.

"4) Ambiguous formulation regarding the sign of J: Below Equation (6), the text states “For F (resp. AF) interactions (J>0)...” This is ambiguous and potentially confusing. I suggest rewriting it more explicitly as: “ferromagnetic (J>0), antiferromagnetic (J<0).” 5) Figure 2 caption: The caption mentions “blue links,” but no blue edges are visible in the figure. Please check whether this is a leftover from an earlier version, or clarify what is meant. 6) Figure 5 caption: In the last sentence, the caption refers to lattice “A3334,” which does not appear elsewhere in the paper. This seems to be a typographical error; presumably it should read “A33344.” 7) Punctuation after equations: The manuscript contains several cases of inconsistent punctuation following displayed equations—sometimes a period or comma is missing, other times a full stop is used even though the sentence continues. I recommend reviewing the manuscript for consistent punctuation to improve clarity and readability. 8) Sentences beginning with variables: In many places, sentences begin with expressions such as f(T), c_v , etc., which visually appear to begin with a lowercase letter. While this is not technically incorrect in scientific writing, I suggest rephrasing such sentences (e.g., “The function f(T)...”) to ensure smoother readability and presentation."

Remarks 4 to 8 have been taken into account.

"9) Applicability to other lattices – open question: It would be valuable if the authors briefly commented in the conclusion on the potential applicability of their approach beyond Archimedean and Laves lattices. For example, can similar methods be applied to other periodic planar graphs with non-uniform vertex degree? Is constant coordination number a necessary condition? Even a short discussion on this point would benefit readers considering extensions of this work."

Similar calculations of the A and B quantities can be derived for planar graphs that are neither Archimedean nor Laves lattices, and that have several values of J on the links in a unit cell. We have precised all along the text the places where the results can easily be extended to more general lattices, and added in the conclusion: "The formulae in this article are directly applicable to any model on a planar lattice with a single type of link, for which the program given in Supp. Mat. [25] can be used. The formulae can be extended when several link types are present, as mentioned in Sec. 2.3 (examples of solutions in [13, 27])."

ANSWER TO REPORT 2

We thank the referee for his/her positive evaluation. We answer below to the requests for minor changes and to typos :

1- Minor changes

"a- References : while the main references are acknowledged and mentioned, a few references are still missing. As they are already mentioned in the other report, let me only add this one : - I.Syôzi, Statistics of Kagomé Lattice, Progress of Theoretical Physics, Volume 6, Issue 3, 1951, Pages 306–308"

This reference has been added, together with all the first references with the first mention of the critical temperature for the Archimedean lattices (in Tab. 1).

"b- Fig. 2 is missing some of the colors mentioned in its caption. I believe these would be extremely helpful to clarify the main text as the results are not completely trivial."

This has been corrected.

2- Typos etc.:

"a- It is unclear to me what u(\check{l}_f) means in Eq. (10). Is this correct ?"

There was some ambiguity in the notations, as \check{l_f} could refer to the check applied to the site l_f or the link l. We thank the referee for seeing it. To avoid this, we do no more use this notation for sites, and only use it for links. Moreover, a picture has been added illustrating the meaning of this notation (fig.2).

"b- The reference to Eq. (19) at the beginning of paragraph (2.3) seems a bit strange (albeit perhaps correct) since it is only presented much later on in the paper."

This has been corrected, and now the reference is to Eq. (14).

"c- Above Eq. (18), I suspect P_0(1) should either have an overline or should be set equal to -2^{2 \tilde{m}}. d- In Eq. (18) I have a doubt: should it not be n_{\nu} ?"

This two typos have been corrected.

"e- There is a typo in the residual entropy for A3636 in sec. 5.2 (0.0518 should be 0.5018), as stated correctly in the related table."

Thanks to the referee for seeing this typo. We have corrected it.

"- I believe "insure" should be "ensure" in both cases. - bottom of P. 6 "we fall is the situation"-> "we fall in the situation". "

This has been corrected.

4- Optional changes / questions for the authors' consideration:

"a- The angle \alpha is introduced quite early in the text but mostly used around Eq. (9). It could be helpful to the reader to only introduce it there and to explicitly mention it in Appendix A."

The definition of alpha has been moved just before eq. 9 (now Eq. 10) and a new figure now illustrates it (Fig 2).

"b- The appendices may be easier to navigate if presented in the same order as the text, or alternatively if the choice of their structure is explained in the introduction or at the very beginning of the appendices."

We have reorganized the appendices so that they now appear in the same order as their references in the text.

"c- The various geometrical quantities introduced in the first sections can be hard to follow. A figure summarizing their different roles might be helpful, or an "intuition" behind their impact on the calculations e.g. above Eq. 9 and e.g. in the very beginning of sec. 2.4 when mentioning Eq. (70)."

We have added a figure illustrating several notations (new Fig.2) used in sec. 2.1 and 2.2. In the beginning of Sec. 2.4, the reference to Eq. (70) of the appendix has been removed, and we now precise in this section what is needed for the following, refering to the appendix only for more details. The dual lattices are now defined here, with the associated figure moved here (Fig. 7 is now Fig. 3).

"c2 - The square torus is used to discuss periodic boundary conditions. In numerics, people sometimes use other constructions for tori to respect the lattice symmetries (e.g. in the triangular lattice case). May the author comment on whether this could affect their results ?"

We misused "square torus" and "square flat torus" instead of "flat torus". We fixed it. Actually our tori are always flat, but may be square or not.

"d- The authors comment in the conclusion about the appearance of geometric characteristic playing an important role in the statistical mechanics of the Ising model. I may have missed it : I am curious whether their systematic analysis (see Table 1 & 3) provides further insight on possible systematic correlations between e.g. the value of n_nu and A, B."

We have removed this remark as we didn't found any simple correlations between geometric characteristics and the values of A and B. This remark was more related to the polynomials properties, as for example the multiplicity of the factor (1+v), or n_v appearing in the ground state energy. But we added two remarks in the main text: we recall the already known correlation between the coordination number and the critical temperature: "On this figure, we recover the strong correlation (quasi-linear dependancy) between the coordination number $z$ and the critical temperature $T_c$, which is not specific to the Ising model" (p.12) and emphasize the lack of simple correlations between geometric characteristics and the values of A and B: " Despite strong variations in the parameters z, m, nl , n v (see Tab. 1), the coefficients A and B of the singularity for the F models show no simple correlations with them." (p.16)

e- The authors may consider summarizing in the conclusions the directions they already pointed out where their results may or may not be generalizable, possible references where this might have been considered, and perhaps comment on a few other possible directions. For instance, is there a way to use the combinatorial method to charaterize the magnetization or the correlations (despite the absence of a solution in the presence of a magnetic field), and for instance identify in the AF model the presence of a zero-temperature critical point? Would the approach be applicable to spin glasses (e.g. J.F. Valdés, W. Lebrecht, E.E. Vogel, Physica A: Statistical Mechanics and its Applications, Volume 391, (2012))? Would it be applicable to amorphous lattices ? These last few questions are mostly out of curiosity inspired by this work.

We have added further details in the text on the validity of the formulas for non-Archimedean lattices, and clarified them in the conclusion: "The formulae in this article are directly applicable to any model on a planar lattice with a single type of link, for which the program given in Supp. Mat. [25] can be used. The formulae can be extended when several link types are present, as mentioned in Sec. 2.3 (examples of solutions in [13, 27]). However, extension to magnetization calculations, or to disordered lattices are left for future work."

---

## Editorial Decision

published